# A Unifying Framework for Parallelizing Sequential Models with Linear Dynamical Systems

**Xavier Gonzalez\***                                                        *xavier18@stanford.edu*
*Statistics Department and Wu Tsai Neurosciences Institute*
*Stanford University*

**E. Kelly Buchanan\***                                                *kelly.buchanan@stanford.edu*
*Statistics Department and Wu Tsai Neurosciences Institute*
*Stanford University*

**Hyun Dong Lee**                                                            *hdlee@stanford.edu*
*Computer Science Department and Wu Tsai Neurosciences Institute*
*Stanford University*

**Jerry Weihong Liu**                                                          *jwl50@stanford.edu*
*Institute for Computational and Mathematical Engineering*
*Stanford University*

**Ke Alexander Wang**                                                    *alxwang@cs.stanford.edu*
*Computer Science Department*
*Stanford University*

**David M. Zoltowski**                                                    *dzoltowski@stanford.edu*
*Statistics Department and Wu Tsai Neurosciences Institute*
*Stanford University*

**Leo Kozachkov**                                                             *leokoz8@brown.edu*
*School of Engineering and Carney Institute for Brain Science*
*Brown University*

**Christopher Ré**                                                          *chrismre@stanford.edu*
*Computer Science Department*
*Stanford University*

**Scott W. Linderman**                                                *scott.linderman@stanford.edu*
*Statistics Department and Wu Tsai Neurosciences Institute*
*Stanford University*

**Reviewed on OpenReview:** *https://openreview.net/forum?id=fw6GgAIGur*

## Abstract

Harnessing parallelism in seemingly sequential models is a central challenge for modern machine learning. Several approaches have been proposed for evaluating sequential processes in parallel using iterative fixed-point methods, like Newton, Picard, and Jacobi iterations. In this work, we show that these methods can be understood within a common framework based on linear dynamical systems (LDSs), where different iteration schemes arise naturally as approximate linearizations of a nonlinear recursion. Moreover, we theoretically analyze the rates of convergence of these methods, and we verify the predictions of this theory with several case studies. This unifying framework highlights shared principles behind these techniques and clarifies when particular fixed-point methods are most likely to be effective. By bridging diverse algorithms through the language of LDSs, the framework provides a clearer theoretical foundation for parallelizing sequential models and points toward new opportunities for efficient and scalable computation.

---

\* indicates lead authors. Author contribution statement in Appendix A.

# 1 Introduction

Sequential processes are ubiquitous in machine learning models. Evaluating a recurrent neural network (Goodfellow et al., 2016), sampling a diffusion model (Sohl-Dickstein et al., 2015; Ho et al., 2020; Song et al., 2021b), generating from a deep state space model (Gu et al., 2022; Smith et al., 2023; Orvieto et al., 2023; Gu & Dao, 2024), and unrolling layers of a deep neural network (He et al., 2016; Vaswani et al., 2017) all involve sequential computations. Naively, these sequential computations require time proportional to the sequence length or the network depth and do not take full advantage of hardware accelerators like GPUs and TPUs (Lim et al., 2024). However, for one important class of computations—*linear* recursions or linear dynamical systems (LDSs)—this bottleneck has been overcome using techniques like the parallel scan[1] (Blelloch, 1990; Fatahalian & Olukotun, 2024). Indeed, the parallelizability of linear recursions is key to scalably evaluating many deep state space models (Smith et al., 2023; Gu & Dao, 2024). A natural question is whether other sequential processes in machine learning, such as those with *nonlinear* recursions, can be similarly accelerated.

On first inspection, the parallel scan algorithm does not seem to generalize to nonlinear recursions. The parallel scan applies to linear recursions because the composition of two linear functions remains linear, whereas the composition of two nonlinear functions is generally more complicated. For example, the composition of two quadratic functions is quartic. Nevertheless, recent works have proposed several techniques to parallelize nonlinear recursions, including Jacobi (Song et al., 2021a), Picard (Shih et al., 2023; Lu et al., 2025), Newton (Danieli et al., 2023; Lim et al., 2024) and quasi-Newton (Tang et al., 2024; Gonzalez et al., 2024) iterations. These techniques were originally applied to different machine learning problems, including parallelizing the training of nonlinear RNNs and sampling from diffusion models, and follow different notations and intuitions, which obscures their underlying similarities.

In this paper, we show that Jacobi, Picard, and Newton iterations all parallelize nonlinear recursions over the sequence length by iteratively linearizing the nonlinear recurrence and evaluating the resulting linear dynamical system in parallel. While the connections between Picard and Newton iterations and their convergence rates are well-known in applied mathematics (Ortega & Rheinboldt, 1970), we make them explicit for nonlinear recursions and characterize when each method works and why.

Our contributions include the following:

- A unifying framework casting all four methods (Newton, quasi-Newton, Picard, and Jacobi) as iterative LDS evaluations (Section 2);

- A theoretical analysis showing that the rate of convergence depends on the stability of the resulting LDS and the fidelity of its approximation to the true linearized dynamics (Section 3); and

- Three case studies—the group word problem, evaluating a nonlinear RNN, and sampling from a discretized Langevin diffusion—validating that this analysis predicts which method suits which problem (Section 4).

These contributions unify and clarify several recently proposed methods in the machine learning literature, and they highlight the centrality of linear dynamical systems for parallelizing seemingly sequential processes.

# 2 Unifying fixed-point iterations using linear dynamical systems

We first introduce notation for evaluating a generic sequence model. Let $x_t \in \mathbb{R}^D$ denote the state at time $t$, and let $f_t$ denote the corresponding transition function at that time point. Throughout this paper, we will use $D$ to denote the dimension of the hidden state and $T$ to denote the sequence length.

**Problem Statement (Sequential Evaluation):** Evaluate the sequence $\mathbf{x}_{1:T} = (x_1, x_2, \ldots, x_T)$ starting from $x_0$ via the recursion,

$$x_{t+1} = f_{t+1}(x_t). \tag{1}$$

---

[1]See Appendix B for an introduction to the parallel scan.

We omit input dependencies for simplicity, but note that an input $u_t$ can be incorporated into the definition of the transition function by letting $f_{t+1}(x_t) := f(x_t, u_t)$.

The recurrence described in eq. (1) cannot be evaluated in parallel in its original form because $x_{t+1}$ depends directly on $x_t$, creating a chain of dependencies. As a result, the computation of each state must wait for the previous state to be computed. This approach takes $\mathcal{O}(T)$ time to evaluate the sequence. Moreover, this inherently sequential approach prevents us from fully leveraging modern hardware accelerators, which can dramatically accelerate parallelizable computations. These nonlinear recursions are ubiquitous, appearing, for example, in the denoising pass of a diffusion model, in the forward pass of a nonlinear RNN, or in the recurrence relations in implicit layers and deep equilibrium models.

Fixed-point methods offer a promising alternative: rather than computing the sequence step by step, make an initial guess for the *entire* trajectory. We denote this initial guess by $\mathbf{x}_{1:T}^{(0)}$. We then iteratively refine this guess, operating over the entire sequence length in parallel, denoting the guess after $i$ fixed-point iterations as $\mathbf{x}_{1:T}^{(i)}$. In particular, the current guess at iteration $i$ is further refined by applying a fixed-point operator $\mathcal{A} : \mathbb{R}^{TD} \mapsto \mathbb{R}^{TD}$ (which depends on the functions $f_t$ and the initial condition $x_0$) according to

$$\mathbf{x}_{1:T}^{(i+1)} = \mathcal{A}\left(\mathbf{x}_{1:T}^{(i)}\right). \tag{2}$$

In order to be a fixed-point operator, $\mathcal{A}$ should have a fixed point $\mathbf{x}_{1:T}^{\star}$. That is, $\mathcal{A}$ should satisfy that $\mathbf{x}_{1:T}^{\star} = \mathcal{A}(\mathbf{x}_{1:T}^{\star})$, where $\mathbf{x}_{1:T}^{\star}$ is the unique root of the system of equations given by

$$x_{t+1} - f_{t+1}(x_t) = 0, \quad \forall t \in \{0, \ldots, T-1\}. \tag{3}$$

A stopping criterion is used to determine when the fixed point iterations in eq. (2) have converged up to some level of desired numerical accuracy.

Many fixed-point operators can be constructed to satisfy this constraint. However, in the context of parallel evaluation of sequences, we can often be much more specific than a generic operator $\mathcal{A}$. In fact, for the four fixed-point methods we consider in this work, the fixed-point operators $\mathcal{A}$ solve a linear time-varying system over the sequence length, with the common form,

$$x_{t+1}^{(i+1)} = f_{t+1}(x_t^{(i)}) + \tilde{A}_{t+1}(x_t^{(i+1)} - x_t^{(i)}) \tag{4}$$

where the transition matrix $\tilde{A}_{t+1} \in \mathbb{R}^{D \times D}$ is determined by the dynamics functions $f_{t+1}$ and the current guess for the state $x_t^{(i)}$. Importantly, eq. (4) is a linear dynamical system for the as yet unknown $\mathbf{x}_{1:T}^{(i+1)}$ in terms of the currently known $\mathbf{x}_{1:T}^{(i)}$, as $x_{t+1}^{(i+1)}$ is a linear function of $x_t^{(i+1)}$ plus a bias $b_{t+1} = f_{t+1}(x_t^{(i)}) - \tilde{A}_{t+1}x_t^{(i)}$, which only depends on the states from the previous fixed-point iteration.

The transition matrix $\tilde{A}_{t+1}$ can be thought of as an approximation to the Jacobian of the dynamics function $\partial f_{t+1}/\partial x_t$. Different fixed-point methods simply use different approaches to linearizing the dynamics function, as shown in Table 1.

Importantly, because the recursion in eq. (4) is an LDS, it can be evaluated with a *parallel scan* (Blelloch, 1990; Nguyen, 2007). The parallel scan is a core primitive that allows the unrolling of $T$ steps from an LDS, $x_{t+1} = A_{t+1}x_t + b_{t+1}$, in $\mathcal{O}(\log T)$ time on a machine with $\mathcal{O}(T)$ processors. The logarithmic time in the sequence length comes from a divide-and-conquer approach made possible from the fact that composition of affine functions is *closed*, i.e. another affine function. To show the closure of composition of affine functions more explicitly, note that if $f_t(x) = A_t x + b_t$

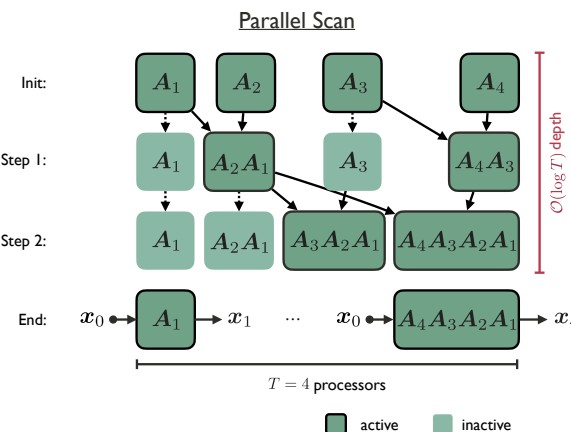

Figure 1: **Parallel scan for LDS.** We illustrate the parallel scan for a sequence of length $T = 4$ for the simple LDS $x_t = A_t x_{t-1}$.

Table 1: **Summary of fixed-point iteration schemes as linear dynamical systems**. We list the methods by the order of their approximation. While higher order methods may converge in fewer iterations, each iteration may be more costly. For example, the prefix sum and parallel scan have $\mathcal{O}(\log T)$ depth, while a single Jacobi iteration has constant depth. For all the methods, each iteration is an LDS, i.e. they can be written in the form of eq. (4) where $\tilde{A}_{t+1}$ is the transition matrix. These methods are guaranteed to converge in at most $T$ iterations (Gonzalez et al., 2024, Proposition 1). "Order" means the highest number of derivatives taken: Newton and quasi-Newton methods use first derivatives, while Picard and Jacobi methods do not use derivatives of $f_t$.

| Fixed-point method | Order | Transition matrix $\tilde{A}_{t+1}$ | Parallelization |
|---|---|---|---|
| Newton | first-order | $\dfrac{\partial f_{t+1}}{\partial x_t}(x_t^{(i)})$ | Parallel Scan (dense matrix multiplication) |
| Quasi-Newton | quasi first-order | $\text{diag}\left[\dfrac{\partial f_{t+1}}{\partial x_t}(x_t^{(i)})\right]$ | Parallel Scan (elementwise vector multiplication) |
| Picard | zeroth-order | $I_D$ | Prefix Sum (vector addition) |
| Jacobi | zeroth-order | $0$ | Map (embarrassingly parallel) |

---

**Algorithm 1** Fixed-point methods for evaluating sequences using LDSs and parallel scan

---

> **procedure** PARALLELFIXEDPOINT($f$, $x_0$, initial guess $\mathbf{x}_{1:T}^{(0)}$, tolerance $\epsilon$)
>> **for** $i = 0, 1, \dots, T$ **do**
>>> $\tilde{A}_{1:T} \leftarrow$ LINEARIZEDYNAMICS($f, x_0, \mathbf{x}_{1:T}^{(i)}$)          $\triangleright$ For all $t$ in parallel
>>> $\mathbf{x}_{1:T}^{(i+1)} \leftarrow$ EVALUATELDS($x_0, \tilde{A}_{1:T}, f, \mathbf{x}_{1:T}^{(i)}$)          $\triangleright$ Using parallel scan
>>> **if** COMPUTEERROR($x_0, \mathbf{x}_{1:T}^{(i+1)}, f$) $< \epsilon$ **then**
>>>> **break**
>> **return** $\mathbf{x}_{1:T}^{(i+1)}$

---

and $f_{t+1}(x) = A_{t+1}x + b_{t+1}$, then $f_{t+1}(f_t(x)) = A_{t+1}A_t x + (A_{t+1}b_t + b_{t+1})$. Therefore, as shown in Figure 1, by combining neighboring affine functions together, we obtain transition maps from the initial condition $x_0$ to all other states $x_t$ in $\mathcal{O}(\log T)$ time. We provide further discussion of the parallel scan algorithm for the interested reader in Appendix B.

Next, we discuss in more detail how the four prominent fixed-point methods discussed in Table 1 reduce to iterative application of LDSs when used to solve a recursion. Fundamentally, all of these methods parallelize nonlinear recursions by iteratively linearizing and evaluating them, as we indicate in Algorithm 1. However, the different methods use different approximations of the Jacobian $\partial f_{t+1}/\partial x_t$ of the dynamics function for their transition matrices $\tilde{A}_{t+1}$.

## 2.1 Newton iterations

Bellen & Zennaro (1989), Gander & Vandewalle (2007), Danieli et al. (2023), and Lim et al. (2024) demonstrated that when the fixed-point operator $\mathcal{A}$ is constructed as an appropriately designed LDS, built from a linearization of original nonlinear recursion $f$, then each application of $\mathcal{A}$ is equivalent to an iteration of *Newton's root-finding method* on the system of equations given in eq. (3). Specifically, each Newton fixed-point iteration, $\mathbf{x}_{1:T}^{(i+1)} = \mathcal{A}_\text{N}\big(\mathbf{x}_{1:T}^{(i)}\big)$, is defined by the linear recursion,

$$x_{t+1}^{(i+1)} = f_{t+1}(x_t^{(i)}) + \frac{\partial f_{t+1}}{\partial x_t}(x_t^{(i)})\left(x_t^{(i+1)} - x_t^{(i)}\right), \tag{5}$$

which we recognize as the first-order Taylor expansion of the nonlinear recursion. Because this fixed-point iteration uses a first derivative, we refer to it as a *first-order* fixed-point iteration. We note that for Newton iterations, the transition matrix is exactly the Jacobian of the dynamics function, $\tilde{A}_{t+1} := \frac{\partial f_{t+1}}{\partial x_t}(x_t^{(i)})$.

Crucially, because eq. (5) is a linear dynamical system, it can be evaluated in parallel over the sequence length using an associative scan (Blelloch, 1990). However, each iteration requires $\mathcal{O}(TD^2)$ memory to store the $T$ Jacobian matrices, and $\mathcal{O}(TD^3)$ work to compute the matrix-matrix multiplications in the parallel scan. This expense is prohibitive for large state size or sequence length, which motivates the quasi-Newton iterations we discuss next.

## 2.2 Quasi-Newton iterations

The compute and memory costs of full Newton iterations have motivated a wide literature on quasi-Newton methods (Nocedal & Wright, 2006). A particularly simple way to turn the Newton iteration in eq. (5) into a quasi-Newton iteration that uses a parallel associative scan was proposed by Gonzalez et al. (2024): just use the diagonal of the Jacobian of the dynamics function. Specifically, each of these quasi-Newton fixed-point iterations $\mathcal{A}_{\text{QN}}$ is given by

$$x_{t+1}^{(i+1)} = f_{t+1}(x_t^{(i)}) + \text{diag}\left[\frac{\partial f_{t+1}}{\partial x_t}(x_t^{(i)})\right]\left(x_t^{(i+1)} - x_t^{(i)}\right), \tag{6}$$

which we recognize as an LDS with transition matrix $\tilde{A}_{t+1} = \text{diag}\left[\frac{\partial f_{t+1}}{\partial x}(x_t^{(i)})\right]$.

With this transition matrix, the parallel scan requires only $\mathcal{O}(TD)$ memory and $\mathcal{O}(TD)$ work, and is therefore more computationally efficient than the full Newton iteration eq. (5). However, quasi-Newton methods may take more fixed-point iterations to converge. As we will show in Sections 3 and 4, whether Newton and quasi-Newton methods are faster for evaluating a given sequence depends on many factors including the accuracy of the approximate Jacobian, the stability of the linearized system, the choice of hardware, and the scale of the problem. In many cases, the large memory consumption of full Newton iterations renders it infeasible for large-scale problems.

Another consideration is the manner in which $\tilde{A}_{t+1}$ is computed. A primary goal of quasi-Newton methods (Broyden, 1970; Dennis Jr & Schnabel, 1996) is to avoid computation of $\partial f_{t+1}/\partial x_t$. Simply computing these Jacobians with autodifferentiation and then taking their diagonal still requires $D$ function calls, which can be costly. However, in sequence modeling with recurrent neural networks (RNNs), the diagonal elements of the Jacobian can often be written in closed form, allowing for its evaluation in a single function call. This approach is taken in Gonzalez et al. (2024) and Danieli et al. (2026). Another approach proposed in Zoltowski et al. (2025) is to stochastically estimate the diagonal using the Hutchinson estimator. This stochastic approach also requires only one function call but is generally applicable even when closed forms of $\tilde{A}_{t+1}$ are not available. We employ this efficient stochastic estimator in all of our experiments. Finally, a third approach comes from the multisecant or Broyden family of methods (Fang & Saad, 2009; Walker & Ni, 2011), including Anderson acceleration (Anderson, 1965; Bai et al., 2019; Tang et al., 2024), which build up estimates of derivative information over the fixed-point iterations.

In general, any structured approximation of the Jacobian matrix that remains closed under matrix multiplication[2] could be used to form a quasi-Newton fixed-point iteration amenable to a parallel scan. For example, Zoltowski et al. (2025) introduces a parallel quasi-Newton method where each block of the matrix is diagonal. The broader development of quasi-Newton methods that fit into the unifying LDS framework discussed in this paper is an important direction for future work, which we elaborate on in Section 6. However, for simplicity, henceforth in this paper when we refer to "quasi-Newton iterations," we restrict ourselves to the simple diagonal approximation shown in eq. (6).

---

[2]Closed in the sense that the product, $A_{t+1}A_t$, has the same structure as $A_{t+1}$ and $A_t$, as with diagonal matrices. See Appendix B for more detail.

### 2.3 Picard iterations

A seemingly different approach to using fixed-point iterations is Picard iterations, which were used by Shih et al. (2023) to parallelize sampling in diffusion models. Picard iterations are often used in the context of evaluating ODEs, where

$$\dot{x} = g(x, t).$$

After Euler discretization with step size $\Delta$, we obtain the discrete-time recursion,

$$x_{t+1} = x_t + g(x_t, t) \cdot \Delta. \tag{7}$$

The Picard fixed-point iteration, $\mathbf{x}_{1:T}^{(i+1)} = \mathcal{A}_P(\mathbf{x}_{1:T}^{(i)})$, is then given by,

$$x_{t+1}^{(i+1)} = x_0 + \sum_{s=0}^{t} g(x_s^{(i)}, s) \cdot \Delta. \tag{8}$$

Because Picard iterations do not use any derivatives of the discrete-time recursion, we call them *zeroth-order* fixed-point iterations.

Shih et al. (2023) proves by induction that for any dynamical system given by eq. (7), the fixed-point iterations given by eq. (8) will converge to the true trajectory in at most $T$ iterations. Similarly, Gonzalez et al. (2024) proved that for any dynamical system given by eq. (1), both the Newton fixed-point iterations given by eq. (5) and the quasi-Newton fixed-point iterations given by eq. (6) will converge to the true trajectory in at most $T$ iterations. The similarity of these results and techniques begs the question as to how Picard and Newton iterations relate to each other. Our first result shows that Picard iterations are in fact a type of quasi-Newton iteration, where we approximate the Jacobian of the dynamics function by the identity matrix.

**Proposition 1.** *The Picard iteration operator $\mathcal{A}_P$ given by eq. (8) is a special case of an LDS, eq. (4), where the transition matrix is the identity,*

$$\tilde{A}_{t+1} = I_D.$$

*Proof.* Define $f_{t+1}(x_t) := x_t + g(x_t, t) \cdot \Delta$. Then, from eq. (8) it follows that

$$\begin{aligned}
x_{t+1}^{(i+1)} &= x_t^{(i+1)} + g(x_t^{(i)}, t) \cdot \Delta \\
&= x_t^{(i+1)} - x_t^{(i)} + x_t^{(i)} + g(x_t^{(i)}, t) \cdot \Delta \\
&= f_{t+1}(x_t^{(i)}) + (x_t^{(i+1)} - x_t^{(i)}).
\end{aligned}$$

This is exactly of the form of the generic linear recursion shown in eq. (4), with $\tilde{A}_{t+1} = I_D$. $\square$

An important consequence of Proposition 1 is that like Newton iterations and quasi-Newton iterations, Picard iterations can also be cast as an LDS. In Newton iterations, the full Jacobian $\partial f_{t+1}/\partial x_t$ is used in LDS; in quasi-Newton iterations, the diagonal approximation $\text{diag}[\partial f_{t+1}/\partial x_t]$ is used; and in Picard iterations, the identity $I_D$ is used. The Picard iteration is more compute and memory efficient than even quasi-Newton, requiring only vector additions via the prefix sum algorithm. However, it is generally a less faithful approximation and takes more iterations to converge, unless the Jacobian is well-approximated by the identity.

### 2.4 Jacobi iterations

Yet another seemingly different class of fixed-point methods are Jacobi iterations (Ortega & Rheinboldt, 1970), which were used by Song et al. (2021a) to accelerate computation in a variety of settings in machine learning, such as feedforward networks with skip connections. Jacobi iterations are also a zeroth-order fixed-point method, and are commonly used to solve systems of multivariate nonlinear equations of the form,

$$h_t(\mathbf{x}_{1:T}) = 0 \quad \forall t \in \{1, \ldots, T\}.$$

Instead, the Jacobi fixed-point operator, $\mathbf{x}_{1:T}^{(i+1)} = \mathcal{A}_J(\mathbf{x}_{1:T}^{(i)})$, solves the following system of $T$ *univariate* equations *in parallel* to obtain $\mathbf{x}_{1:T}^{(i+1)}$,

$$h_t^{(i)}\left(x_1^{(i)}, \ldots, x_{t-1}^{(i)}, x_t, x_{t+1}^{(i)}, \ldots, x_T^{(i)}\right) = 0 \quad \forall t \in \{1, \ldots, T\} \tag{9}$$

Song et al. (2021a) considers in particular the problem of solving recurrence relations of the form $x_{t+1} = f_{t+1}(\mathbf{x}_{1:t})$, and proves that, for such a system, Jacobi iterations converge in at most $T$ iterations. This result is directly analogous to similar such results and proofs in Gonzalez et al. (2024) and Tang et al. (2024) for Newton and quasi-Newton iterations and Shih et al. (2023) for Picard iterations. In fact, in the context of iteratively applying LDSs to parallelize Markovian state space models, we prove that Jacobi iterations are a type of degenerate quasi-Newton iterations, where we "approximate" the Jacobian of the dynamics function by zero.

**Proposition 2.** *When applied to a Markovian state space model as in eq.* (1), *the Jacobi iteration operator $\mathcal{A}_J$ specified by eq.* (9) *is a special case of the common form, eq.* (4), *where,*

$$\tilde{A}_{t+1} = 0.$$

*Proof.* In a Markovian state space model, the recurrence relation always takes the form specified in eq. (1), i.e. $x_{t+1} = f_{t+1}(x_t)$. Thus, Jacobi iterations take the simple form

$$x_{t+1}^{(i+1)} = f_{t+1}(x_t^{(i)}).$$

Because $x_{t+1}^{(i+1)}$ does not depend on $x_t^{(i+1)}$, we see that the transition matrix is zero. $\square$

We have shown that evaluating nonlinear recursions via Newton, quasi-Newton, Picard, and Jacobi fixed-point iterations reduces to iteratively solving linear dynamical systems. An important corollary is that these methods are guaranteed to converge in all problem settings in at most $T$ iterations (Gonzalez et al., 2024, Prop. 1). However, the precise convergence rates of the different fixed-point methods will depend on the accuracy of the Jacobian approximation, as we show next.

## 3 Analysis of convergence rates

In this section, we analyze the convergence properties of the fixed-point methods above. We show that the convergence rate of these fixed-point methods depends on two factors: (i) how well the transition matrix $\tilde{A}_{t+1}$ approximates the true dynamics Jacobian $A_{t+1} := \partial f_{t+1}/\partial x_t$; and (ii) the stability of the LDS with transition matrices $\tilde{A}_{1:T}$.

### 3.1 Derivation of fixed-point iterations from high-dimensional root-finding.

We first review the derivation of the Newton fixed-point iterations in eq. (5) from the system of equations given by eq. (3), highlighting how the other fixed-point iterations naturally arise from different approximations of $\partial f_{t+1}/\partial x_t$. This background will be useful for proving convergence rates in Section 3.2.

Returning to the system of nonlinear equations in eq. (3), we define the residual function $\mathbf{r} : \mathbb{R}^{TD} \mapsto \mathbb{R}^{TD}$,

$$\mathbf{r}(\mathbf{x}_{1:T}) = [x_1 - f_1(x_0), \ x_2 - f_2(x_1), \ x_3 - f_3(x_2), \ldots, x_T - f_T(x_{T-1})]. \tag{10}$$

We define $\mathbf{J}(\mathbf{x}_{1:T}) \in \mathbb{R}^{TD \times TD}$ to be the Jacobian of the residual $\mathbf{r}(\mathbf{x}_{1:T})$,

$$\mathbf{J}(\mathbf{x}_{1:T}) := \frac{\partial \mathbf{r}}{\partial \mathbf{x}_{1:T}}(\mathbf{x}_{1:T}) = \begin{pmatrix} I_D & 0 & 0 & \ldots & 0 & 0 \\ -\frac{\partial f_2}{\partial x}(x_1) & I_D & 0 & \ldots & 0 & 0 \\ 0 & -\frac{\partial f_3}{\partial x}(x_2) & I_D & \ldots & 0 & 0 \\ \vdots & \vdots & \vdots & \ddots & \vdots & \vdots \\ 0 & 0 & 0 & \ldots & I_D & 0 \\ 0 & 0 & 0 & \ldots & -\frac{\partial f_T}{\partial x}(x_{T-1}) & I_D \end{pmatrix}. \tag{11}$$

Therefore, in this notation, every Newton's method update takes the form

$$\mathbf{x}_{1:T}^{(i+1)} = \mathcal{A}_N(\mathbf{x}_{1:T}^{(i)}) := \mathbf{x}_{1:T}^{(i)} - \mathbf{J}(\mathbf{x}_{1:T}^{(i)})^{-1}\mathbf{r}(\mathbf{x}_{1:T}^{(i)}).$$

Using matrix manipulations, it can be shown (Danieli et al., 2023; Lim et al., 2024; Gonzalez et al., 2024) that each Newton fixed-point iteration is given by the linear recursion in eq. (5).

Analogously, for the other fixed point iterations considered in this paper (cf. Table 1), which use transition matrices $\tilde{A}_{t+1}$ to approximate $\partial f_{t+1}/\partial x_t$, we define a block-bidiagonal $\widetilde{\mathbf{J}} \in \mathbb{R}^{TD \times TD}$ simply by substituting $\tilde{A}_{t+1}(x_t)$ for $\partial f_{t+1}/\partial x_t$ in eq. (11). The corresponding fixed point iteration $\mathcal{A}$ takes the form

$$\mathcal{A}(\mathbf{x}_{1:T}^{(i)}) := \mathbf{x}_{1:T}^{(i)} - \widetilde{\mathbf{J}}(\mathbf{x}_{1:T}^{(i)})^{-1}\mathbf{r}(\mathbf{x}_{1:T}^{(i)}). \tag{12}$$

Different fixed-point methods $\mathcal{A}$ give rise to different matrices $\widetilde{\mathbf{J}}$, which impacts their convergence rates.

## 3.2 Convergence rates of fixed-point iterations

We derive convergence rates for all fixed-point operators discussed in this paper. The analysis follows standard techniques in the literature (cf. Kelley (1995, Thm 5.4.1)), which Lu et al. (2025, Prop 4.) applied to parallelizing sequential models with Picard iterations. Our unifying framework allows us to see that their analysis generalizes straightforwardly.

Let $\mathbf{e}_{1:T}^{(i)} := \mathbf{x}_{1:T}^{(i)} - \mathbf{x}_{1:T}^{\star} \in \mathbb{R}^{TD}$ denote the error of a trajectory $\mathbf{x}_{1:T}^{(i)}$. Note that the error depends on the current trajectory and the dynamics functions, but we suppress these dependencies to keep notation lightweight. For any of the fixed-point methods discussed in this paper, we can bound the convergence rate of this error.

**Proposition 3** (c.f. Proposition 4 of Lu et al. (2025)). *Consider a fixed-point solver with updates given by eq. (12) for some matrix $\widetilde{\mathbf{J}}(\mathbf{x}_{1:T}^{(i)})$ with form specified by eq. (19). Let $L$ be the maximum of the Lipschitz constants of $\partial f_t/\partial x_{t-1}$. Then $\|\mathbf{e}_{1:T}^{(i+1)}\|_2$ satisfies*

$$\|\mathbf{e}_{1:T}^{(i+1)}\|_2 \le \left\|\widetilde{\mathbf{J}}(\mathbf{x}_{1:T}^{(i)})^{-1}\right\|_2 \cdot \left(\left\|\widetilde{\mathbf{J}}(\mathbf{x}_{1:T}^{(i)}) - \mathbf{J}(\mathbf{x}_{1:T}^{(i)})\right\|_2 \|\mathbf{e}_{1:T}^{(i)}\|_2 + \frac{L}{2}(\|\mathbf{e}_{1:T}^{(i)}\|_2^2)\right), \tag{13}$$

*where $\|\cdot\|_2$ denotes the spectral norm of a matrix and the $\ell_2$ norm of a vector.*

*Proof.* Starting from eq. (12), we subtract $\mathbf{x}_{1:T}^{\star}$ from both sides to obtain

$$\mathbf{e}_{1:T}^{(i+1)} = \mathbf{e}_{1:T}^{(i)} - \widetilde{\mathbf{J}}(\mathbf{x}_{1:T}^{(i)})^{-1}\mathbf{r}(\mathbf{x}_{1:T}^{(i)}).$$

Next, we Taylor expand $\mathbf{r}(\cdot)$ around $\mathbf{x}_{1:T}^{(i)}$ to obtain

$$\mathbf{r}(\mathbf{x}_{1:T}^{\star}) = \mathbf{r}(\mathbf{x}_{1:T}^{(i)}) - \mathbf{J}(\mathbf{x}_{1:T}^{(i)})\mathbf{e}_{1:T}^{(i)} + \mathbf{R}(\mathbf{e}_{1:T}^{(i)}),$$

where $\mathbf{R}(\mathbf{e}_{1:T}^{(i)})$ is the second-order remainder function and has norm bounded by $\|\mathbf{e}_{1:T}^{(i)}\|_2^2/2$ times the Lipschitz constant of $\mathbf{J}(\mathbf{x}_{1:T}^{(i)})$, which Theorem 3 of Gonzalez et al. (2025) shows is bounded by $L$. Since $\mathbf{r}(\mathbf{x}_{1:T}^{\star}) = \mathbf{0}$, it follows that

$$\mathbf{e}_{1:T}^{(i+1)} = \widetilde{\mathbf{J}}(\mathbf{x}_{1:T}^{(i)})^{-1}\left(\left(\widetilde{\mathbf{J}}(\mathbf{x}_{1:T}^{(i)}) - \mathbf{J}(\mathbf{x}_{1:T}^{(i)})\right)\mathbf{e}_{1:T}^{(i)} + \mathbf{R}(\mathbf{e}_{1:T}^{(i)})\right). \tag{14}$$

The result follows by taking norms on both sides and using the triangle inequality. $\square$

## 3.3 Intuitions about rates of convergence

Proposition 3 shows that, to first order, the error is determined by the discrepancy between the chosen linear operator $\widetilde{\mathbf{J}}$ and the true Jacobian $\mathbf{J}$ of the residual. Moreover, we see that as $\|\mathbf{e}_{1:T}^{(i)}\|_2$ approaches zero, the

first-order term, which is linear in $\|\mathbf{e}_{1:T}^{(i)}\|_2$, must eventually (under strong enough continuity assumptions) dominate the second-order term, which is quadratic in $\|\mathbf{e}_{1:T}^{(i)}\|_2$. Typically, we would say the rate of decrease in $\|\mathbf{e}_{1:T}^{(i)}\|_2$ approaches an asymptotic linear rate $\gamma$ given by

$$\gamma := \left\|\widetilde{\mathbf{J}}(\mathbf{x}_{1:T}^\star)^{-1}\right\|_2 \left\|\widetilde{\mathbf{J}}(\mathbf{x}_{1:T}^\star) - \mathbf{J}(\mathbf{x}_{1:T}^\star)\right\|_2. \tag{15}$$

Discussions of asymptotic linear rate are subtle in our setting, where all fixed-point methods are guaranteed to converge in $T$ iterations (see our discussion in Appendix F.1). Nonetheless, the functional form of $\gamma$ provides useful intuition about the convergence rates of different fixed-point methods.

Let's study the two factors in the asymptotic linear rate $\gamma$, which must be less than one to be meaningful. Lu et al. (2025) left quantifying these two factors for future work, which we now bring to fruition. The first factor is the spectral norm of the inverse of the approximate Jacobian. Because $\widetilde{\mathbf{J}}(\mathbf{x}_{1:T})$ is a block bidiagonal matrix (see eq. (11) and the definition that follows), its inverse has a block lower triangular structure of the form

$$\widetilde{\mathbf{J}}(\mathbf{x}_{1:4})^{-1} = \begin{pmatrix} I_D & 0 & 0 & 0 \\ \tilde{A}_2 & I_D & 0 & 0 \\ \tilde{A}_3\tilde{A}_2 & \tilde{A}_3 & I_D & 0 \\ \tilde{A}_4\tilde{A}_3\tilde{A}_2 & \tilde{A}_4\tilde{A}_3 & \tilde{A}_4 & I_D \end{pmatrix}, \tag{16}$$

shown above for $T = 4$.

From eq. (16), we see that the blocks of $\widetilde{\mathbf{J}}(\mathbf{x}_{1:T})^{-1}$ are products of the transition matrices $\tilde{A}_{t+1}$ from the chosen fixed-point method. In particular, if the chosen fixed point method results in an *unstable* LDS with $\|\tilde{A}_{t+1}\|_2 > 1$ for many time points, the spectral norm of $\widetilde{\mathbf{J}}(\mathbf{x}_{1:T})^{-1}$ can be much larger than one. Intuitively, fixed-point methods resulting in unstable LDSs will have slower rates of convergence since instability leads to numerical divergence. We elaborate on this point in Appendix C.3.2.

The second factor is the spectral norm of the difference between the approximate and true *residual* Jacobian. The following lemma shows that we can control this factor in terms of the spectral norms of the differences between the approximate and true *dynamics* Jacobians. Intuitively, the rate of convergence will be faster if the transition matrices $\tilde{A}_{t+1}$ are closer to the true dynamics Jacobian $A_{t+1} := \partial f_{t+1}/\partial x_t$ in spectral norm.

**Lemma 1.** *If $\widetilde{\mathbf{J}}(\mathbf{x}_{1:T})$ is given by eq. (19) and $\mathbf{J}(\mathbf{x}_{1:T})$ is given by eq. (11), then*

$$\mathrm{diff}(\mathcal{A}) := \left\|\widetilde{\mathbf{J}}(\mathbf{x}_{1:T}) - \mathbf{J}(\mathbf{x}_{1:T})\right\|_2 = \max_{1 \le t \le T-1} \left\|\tilde{A}_{t+1}(x_t) - A_{t+1}(x_t)\right\|_2,$$

*where $A_{t+1} := \frac{\partial f_{t+1}}{\partial x_t}(x_t)$.*

*Proof.* Appendix C.3.1. □

Proposition 3 is an upper bound on the norm of the error of each fixed-point iterate. As an upper bound, this result can not always fully predict the precise trajectory of the norm of the error (see Appendix C.2). Nevertheless, this proposition provides helpful intuitions that are borne out in the case studies below.

## 4 Case studies: performance of the different fixed-point methods

In this section, we consider three empirical case studies that illustrate how the unifying framework and convergence analysis presented in this paper provides guidance about which fixed-point schemes will excel in which settings. This concordance is based on the structure of the Jacobian of $f_{t+1}$ and the relative computational cost of different fixed-point methods. In a nutshell, we advise:

*Use the simplest approximate Jacobian as possible, but no simpler.*

To elaborate: simpler approximate Jacobians are less computationally expensive, meaning that each fixed-point iteration is more efficient. So, if a lower-order fixed-point method (e.g., Picard or Jacobi) converges in

a small number of iterations, it may reach $\mathbf{x}^\star$ in less time than higher-order methods. However, if the higher-order fixed-point method (e.g. Newton or quasi-Newton) converges in far fewer iterations than lower-order methods, then the increased computation of the higher-order fixed-point method may be worthwhile. As supported by the theoretical analysis in Section 3, the number of iterations needed for a fixed-point method to converge is related to the difference in spectral norm between $\tilde{A}_t$ and $A_t := \partial f_{t+1}/\partial x_t$. We support this intuition with the following case studies, with further experimental details provided in Appendix E. Our code is available at `https://github.com/lindermanlab/parallelizing_with_lds`

## 4.1 Case study #1: Solving the group word problem with Newton iterations

Newton iterations should outperform quasi-Newton and Picard iterations in settings where the Jacobian of the recursion, $f_{t+1}$, is not well approximated by its diagonal, the identity matrix, or the zero matrix. One example of such a recursion is the *group word problem*, which has been used to theoretically and empirically assess the limits of sequential modeling architectures for state-tracking tasks (Kim & Schuster, 2023; Liu et al., 2023; Merrill et al., 2024; Grazzi et al., 2025; Schöne et al., 2025). In the sequence-modeling community, the term "group word problem" is defined as follows.

**Definition 1** (Group Word Problem). *Let $G$ be a finite group and let $g_1, g_2, \ldots, g_T$ be a sequence of group elements. The group word problem is to evaluate the product $g_1 \cdot g_2 \cdots g_T$. Since each $g_t \in G$, the product of these group elements belongs to $G$ as well.*

Merrill et al. (2024) emphasizes that nonlinear RNNs in both theory and practice are able to learn the group word problem in arbitrary groups to high accuracy with only a single layer, whereas compositions of popular linear RNNs linked by nonlinearities, such as S4 (Gu et al., 2022) and Mamba[3] (Gu & Dao, 2024), require a number of layers that grows with $T$. Merrill et al. (2024) emphasizes that recurrent architectures with nonlinear transitions are well-suited for solving the group word problem, because in theory and practice, such architectures can learn the group word problem to high accuracy with a single layer. Other literature has explored the value of matrix-valued states (Beck et al., 2024; Grazzi et al., 2025). However, in Proposition 4 below, we show that neither nonlinearity nor matrix-valued states are needed to understand or solve the group word problem. Instead, the problem can be formulated as an LDS with vector-valued states and input-dependent transition matrices.

**Proposition 4.** *Let $G$ be a finite group. Then there exists some $D \leq |G|$ for which we can represent the group word problem as a time-varying LDS, $f_{t+1}(x_t) = A_{t+1}x_t$, with states $x_t \in \mathbb{R}^D$ denoting the running product of group elements and transition matrices $A_{t+1} \in \mathbb{R}^{D \times D}$ that depend on the input $g_{t+1}$.*

*Proof.* By Cayley's theorem, any finite group $G$ can be embedded in a symmetric group $S_D$, for some $D \leq |G|$. Therefore, by choosing the initial state $x_0 \in \mathbb{R}^D$ to have $D$ distinct entries (a "vocabulary" of size $D$), we can use the tabular representation of permutations (Artin, 2011, eq. 1.5.2) to represent an element of $S_D$ as $x_t$ (by a permutation of the elements of $x_0$). We can also choose $A_{t+1} \in \mathbb{R}^{D \times D}$ to be the permutation matrix corresponding to the embedding of $g_{t+1}$ in $S_D$, since any element of $S_D$ can be represented as a $D \times D$ permutation matrix (e.g., see Figure 2B). Consequently $x_t = A_t A_{t-1} \ldots A_2 A_1 x_0$ is an embedding of an element of $G$ in $S_D$ in the tabular representation. In fact, $x_t \in \mathbb{R}^D$ represents the running product $g_1 g_2 \ldots g_{t-1} g_t$, which is precisely the goal of the group word problem. $\square$

Though we have cast the group word problem as a time-varying LDS with $f_{t+1}(x_t) = A_{t+1}x_t$, we can still evaluate this recursion with any of the fixed-point methods described above. Since the dynamics are linear, the Newton iteration corresponds to evaluating the LDS with a parallel scan, and it converges in one iteration. While other methods would require more iterations to converge, they could still be more efficient in wall-clock time, since they use less memory and compute per iteration. However, we can use the Jacobian approximation error of the different fixed-point methods (Lemma 1) to get a sense if the other fixed-point methods are likely to excel in this setting.

---

[3]Mamba allows input-dependent dynamics matrices but they must be diagonal, which prevents a single Mamba layer from implementing the particular LDS in Proposition 4, which uses permutation matrices. Merrill et al. (2024) also demonstrate that a linear time-varying system with a dense transition matrix can learn the group word problem.

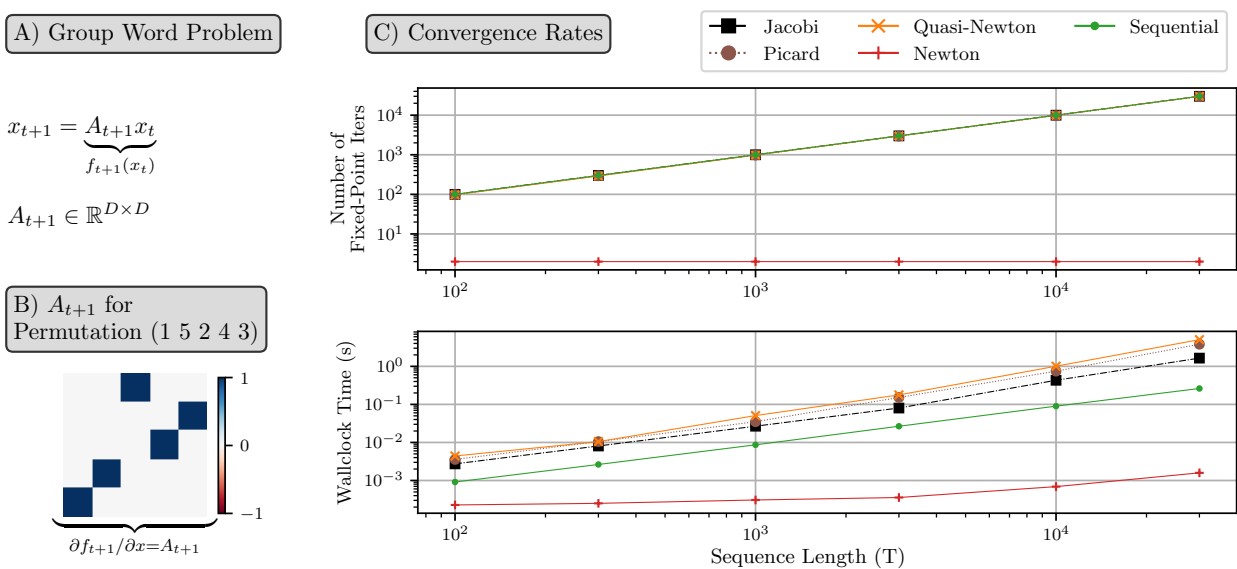

Figure 2: **A single Newton iteration solves the $S_5$ group word problem, whereas the number of iterations required for the other methods increases with sequence length.** We consider the task of evaluating the product of $S_5$ group elements. **A:** The group word problem can be expressed as an LDS with input-dependent state-transition matrices. **B:** An example input-dependent transition matrix $A_t$ for permutation (1 5 2 4 3), in cycle notation. **C:** For each fixed-point method and a range of sequence lengths, $T$, we compute the median (over ten random seeds) number of fixed-point iterations to converge (top) and the median wall-clock time (bottom). While a single Newton iteration is sufficient to solve the $S_5$ problem, the number of iterations required for the other methods increases with the sequence length.

The state transition matrices of the group word problem are permutation matrices with spectral norm one, and so diff$(\mathcal{A}_J) = 1$. Furthermore, since with high probability there will be a state transition matrix with no non-zero entries on the diagonal, it follows that diff$(\mathcal{A}_{QN}) = 1$ while diff$(\mathcal{A}_P) = 2$. Since we would expect to need diff$(\mathcal{A}) < 1$ for a fixed-point method $\mathcal{A}$ to be effective, our theoretical analysis in Section 3 suggests that none of the fixed-point methods other than Newton will be effective on the group word problem.

We test this hypothesis with a simple experiment simulating the $S_5$ word problem, a standard problem in the sequence modeling literature (Merrill et al., 2024; Grazzi et al., 2025). In this setting, Figure 2 shows that quasi-Newton, Picard, and Jacobi iterations require nearly $T$ iterations to converge. On the other hand, we see that Newton's method solves the $S_5$ word problem with just one fixed-point iteration, as expected since the true dynamics are linear. The speed-up is also apparent in the wall-clock time comparison, where we see that Newton is faster than other methods, regardless of $T$.

## 4.2 Case Study #2: Picard iterations struggle to parallelize RNNs

Next, we consider a task where Picard iterations struggle and other fixed-point methods excel, namely, parallelizing recurrent neural networks (RNNs) like the Gated Recurrent Unit (GRU; Cho et al., 2014). We evaluate GRUs with random parameter initialization for different hidden dimension sizes $D$ and sequence lengths $T$ using sequential evaluation as well as fixed-point iterations. Figure 3B shows that at initialization the Jacobian has small entries (on the order of 0.1). In this regime, diff$(\mathcal{A}_J)$ and diff$(\mathcal{A}_{QN})$ are less than one, while diff$(\mathcal{A}_P)$ is greater than one. (See also, Figure 8 in Appendix E.3.) Thus, as expected, Newton, quasi-Newton, and Jacobi iterations excel in this setting, while Picard iterations converge prohibitively slowly, as shown in Figure 3C.

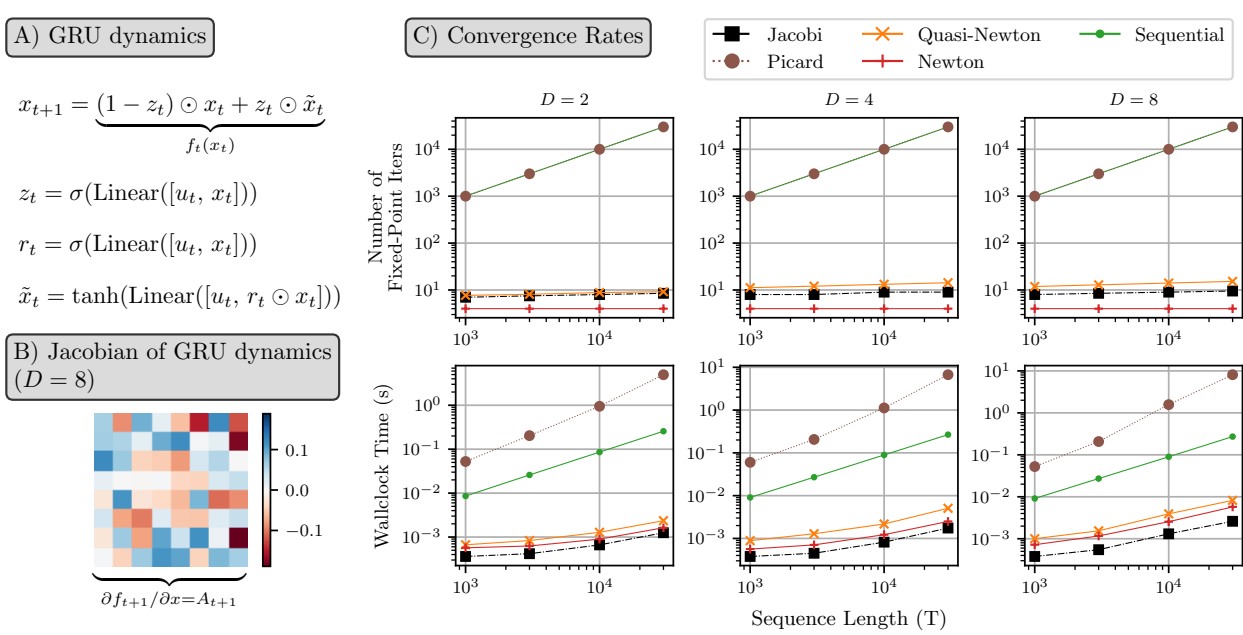

Figure 3: **Picard iterations struggle to parallelize RNNs.** We evaluate GRUs with random parameter initialization for different sequence lengths $T$ and hidden state sizes $D$. **A:** The nonlinear dynamics of a GRU, following Feng et al. (2024), where $x_t$ is the hidden state, $u_t$ is the input, and the notation $\text{Linear}[\cdot, \cdot]$ indicates a linear readout from the concatenation of two vectors. **B:** A representative Jacobian matrix $\partial f_t / \partial x$ from a GRU trajectory, which is not well approximated by the identity matrix. **C:** For each fixed-point method and a range of sequence lengths, $T$, and state sizes, $D$, we compute the median (over ten random seeds) number of fixed-point iterations to converge (top row) and the median wall-clock time (bottom row). Picard iterations take nearly $T$ iterations to converge, while the other fixed point methods yield order-of-magnitude speed-ups over sequential evaluation

### 4.3 Case Study #3: Jacobi iterations struggle to parallelize discretized Langevin diffusion

Based on the theoretical analysis presented in Proposition 3, we expect that if the Jacobian of the dynamics function is well-approximated by the identity matrix, then Picard should converge relatively quickly and at considerably lower cost, especially when compared to the other zeroth-order method of Jacobi iterations. A canonical example of such a system where the dynamics are close to identity comes from a discretization of Langevin dynamics (Langevin, 1908; Friedman, 2022). Langevin dynamics are a workhorse for MCMC (Besag, 1994) and motivated the development of score-matching methods (Song & Ermon, 2019), which are closely related to diffusion models (Sohl-Dickstein et al., 2015; Ho et al., 2020; Song et al., 2021b).

In Langevin dynamics for a potential $\phi$, the state $x_t$ evolves according to a nonlinear recursion with dynamics,

$$f_{t+1}(x_t) = x_t - \epsilon \nabla \phi(x_t) + \sqrt{2\epsilon} w_t,$$

where $\epsilon$ is the step size and $w_t \overset{\text{iid}}{\sim} \mathcal{N}(0, I_D)$. Since the Jacobian is, $\frac{\partial f_{t+1}}{\partial x_t}(x_t) = I_D - \epsilon \nabla^2 \phi(x_t)$, it follows that the identity approximation should work well with small step-sizes $\epsilon$. More generally, the identity approximation tends to be well-suited to problems where a differential equation is discretized with small step sizes, such as when sampling from diffusion models (Holderrieth & Erives, 2025; Lai et al., 2025). In fact, simply by observing the structure of the Jacobian in Panel B of Figure 4, we observe that $\text{diff}(\cdot)$ operator for Newton, quasi-Newton, and Picard iterations in this setting will be close to zero, while $\text{diff}(\mathcal{A}_J)$ will be close to one. Therefore, based on our analysis in Proposition 3, we hypothesize that the other fixed-point methods should dramatically outperform Jacobi iterations in this setting.

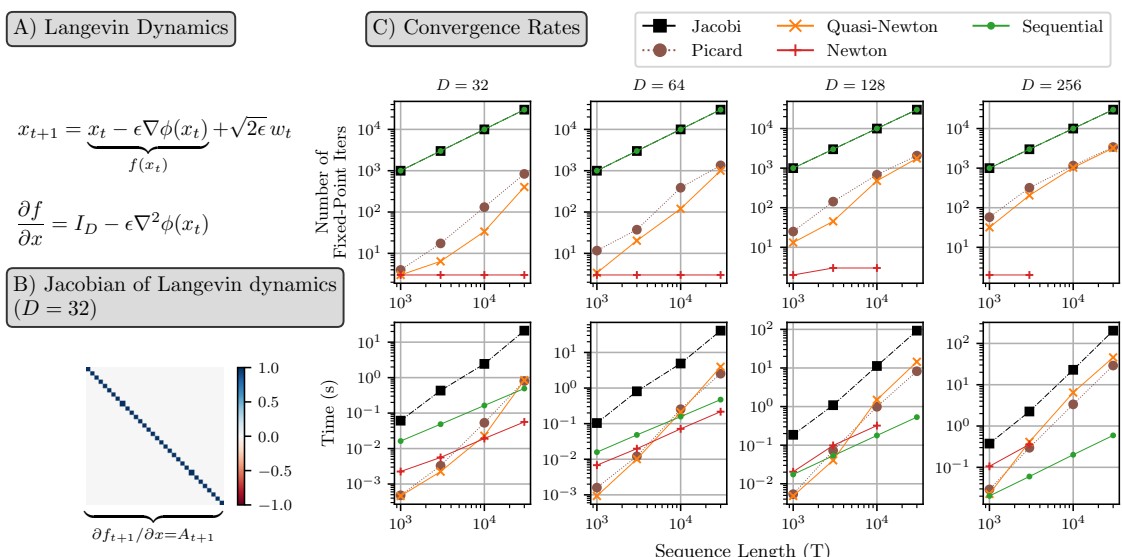

Figure 4: **Jacobi iterations struggle when the dynamics Jacobian is close to the identity.** We evaluate Langevin dynamics for a potential $\phi$. **A:** The nonlinear dynamics of Langevin dynamics for a potential $\phi$ and step size $\epsilon$, where $x_t$ is the state and $w_t$ is Gaussian noise. **B:** The Jacobian for Langevin dynamics is well-approximated by the identity matrix, especially for small step size $\epsilon = 1 \times 10^{-5}$. **C:** We evaluate Langevin dynamics for larger dimensions, plotting the median of 10 random seeds. Jacobi iteration consistently take $T$ steps and are always slower than sequential, while the other fixed-point methods converge in fewer $T$ steps and can be faster than sequential. The missing Newton iteration points indicate the GPU ran out of memory.

We test this hypothesis with a simple experiment shown in Figure 4. We simulate Langevin dynamics on a potential $\phi$ given by the negative log probability of the mixture of two anisotropic Gaussians. In this setting, Picard iterations take far fewer than $T$ iterations to converge and can be faster than sequential evaluation. We note that quasi-Newton iterations, which include information only about the diagonal of the Jacobian of the dynamics, appear to have comparable wall-clock time, by virtue of taking fewer iterations to converge (though each fixed-point iteration involves more work).

Whether fixed-point iterations are faster than sequential evaluation also depends on memory utilization. For example, Shih et al. (2023) and Lu et al. (2025) demonstrated wall-clock speed-ups when using Picard iterations for sampling from a diffusion model using a "sliding window" to only evaluate chunks of the sequence length where the parallel scan algorithm can fit in memory. We discuss these considerations further in Appendix E.5.

## 5 Related Work

In this paper we unify prominent fixed-point methods for the parallel evaluation of sequences in the language of linear dynamical systems. While many papers have employed different fixed-point iterations for different problems in machine learning—Lim et al. (2024), Danieli et al. (2023), and Danieli et al. (2026) using Newton iterations, Tang et al. (2024) and Gonzalez et al. (2024) using quasi-Newton iterations, Shih et al. (2023) using Picard iterations, and Song et al. (2021a) using Jacobi iterations, among other works—to the best of our knowledge no one in the machine learning community has explicitly unified these different methods in the language of linear dynamical systems. It is important to note, however, that Gander & Vandewalle (2007) propose an expansive notion of parareal iterations which include these LDSs as a special case. However, Gander & Vandewalle (2007) focus on solving continuous time differential equations and do not discuss the parallel scan that has become a workhorse for the application of parallel-in-time methods on GPUs in machine learning.

**General unification of fixed-point methods: parallel-chord methods** While connections between Newton's method and Picard iterations have been made before outside of the machine learning literature, our contribution is the tight coupling of these methods to LDSs in the context of parallel evaluation of nonlinear sequences. Ortega & Rheinboldt (1970, Ch. 7) considered the problem of solving a nonlinear equation $\mathbf{F}(\mathbf{x}) = \mathbf{0}$. They showed that Newton and Picard iterations are special cases of general iterative methods where each iterate is given by

$$\mathbf{x}^{(i+1)} = \mathbf{x}^{(i)} - \widetilde{\mathbf{J}}(\mathbf{x}^{(i)})^{-1}\mathbf{F}(\mathbf{x}^{(i)}), \tag{17}$$

for some matrix $\widetilde{\mathbf{J}}(\mathbf{x}^{(i)})$. We discuss the relationship between the unifying frameworks put forward in Ortega & Rheinboldt (1970) and in our paper at greater length in Appendix F.1. The primary difference is that by focusing on the setting of nonlinear sequence evaluation, we bring into greater focus the role of the Jacobian of the dynamics function. Moreover, by unifying fixed-point iterations in the language of LDSs, we emphasize their parallelizability over the sequence length using the parallel scan (Blelloch, 1990).

**Convergence rates of fixed-point methods** In the context of analysis of fixed-point methods in general, there is a broad literature (Ortega & Rheinboldt, 1970; Young, 2014) on the convergence rates of different fixed-point methods. For example, Ortega & Rheinboldt (1970) also proved convergence rates for iterative methods of the form in eq. (17). Though their methods have much in common with the proof techniques used to prove Proposition 3 of this paper, their provided results are actually trivial in the setting considered in this paper. Part of the reason[4] for the inapplicability of the convergence results from Ortega & Rheinboldt (1970) to our paper is that Ortega & Rheinboldt (1970) consider the asymptotic setting, while it has been firmly established that in the particular setting considered in this paper, Jacobi, Picard, quasi-Newton, and Newton iterations all globally converge in at most $T$ iterations (Bellen & Zennaro, 1989; Shih et al., 2023; Tang et al., 2024; Gonzalez et al., 2024). For moving beyond this worst-case analysis, Gonzalez et al. (2025) uses an optimization perspective to show that the difficulty of parallelizing a dynamical system is directly related to the stability of the system, which can be thought of as the "average" spectral norm of $\partial f_{t+1}/\partial x_t$. Proposition 4 of Lu et al. (2025) nicely presents the foundations of the convergence analysis we present in Proposition 3 of this paper. However, we extend their work by applying it to a wider variety of fixed-point methods, explicitly bounding many quantities of interest, and demonstrating its relevance in simulation.

**Other fixed-point methods: mixing sequential with parallel** In this note, we focus on Jacobi, Picard, and Newton iterations because of their prominence (Song et al., 2019; 2021a; Shih et al., 2023; Danieli et al., 2023; Lim et al., 2024; Gonzalez et al., 2024; Grazzi & Zanella, 2025; Zoltowski et al., 2025; Farsang & Grosu, 2025; Danieli et al., 2026; Iacob et al., 2025) and their relationship to LDSs, as listed in Table 1. However, there is a wide literature on iterative solvers (Ortega & Rheinboldt, 1970; Young, 2014). Many of these other methods can also be parallelized over the sequence length, or provide a mixture of parallel and sequential computation. For example, Horton et al. (1995) apply the parallel associative scan (which they call parallel cyclic reduction) for parallel-in-time differential equation solvers, though they focus on Jacobi and Gauss-Seidel iterations. In discrete time, Naumov (2017) shows how evaluating Markov chains can be cast as a system of nonlinear equations and discusses many techniques from numerical analysis for solving them, again focusing on Jacobi and Gauss-Seidel; Song et al. (2021a) extend this program with many deep learning experiments. Although Gauss-Seidel iterations reduce to sequential evaluation when applied to Markovian processes, Song et al. (2021a) also emphasize how the structure of the problem and hardware considerations dictate the optimal mixture of parallel and sequential computation. Parareal iterations (Lions et al., 2001) mix parallel and sequential computation by applying parallelization at multiple length scales, and have also been used to parallelize diffusion models (Selvam et al., 2024). Tang et al. (2024) also parallelize diffusion models using both a generalization of Jacobi iterations, as well as Anderson acceleration (Anderson, 1965; Walker & Ni, 2011), which they modify to be a form of quasi-Newton. We discuss other fixed-point methods based on trust-region techniques in Appendix F.2.

---

[4]We elaborate in Appendix F.1.

## 6 Discussion

This work unified a variety of approaches for parallelizing recursions via fixed-point iterations—including zeroth-order methods like Jacobi and Picard iterations as well as first-order methods like Newton and quasi-Newton iterations—under a common framework. In each case, the iterates reduce to evaluating an appropriately constructed linear dynamical system, which approximates the nonlinear recursion of interest. This unifying framework provides insight into which different problems in machine learning are likely to benefit from which types of fixed-point iterations. In particular, we demonstrate that the structure of the Jacobian matrix of the dynamics function plays a key role in determining which fixed-point method to use.

For this reason, understanding the structure of the Jacobian of the dynamics function is important for using our framework. Fortunately, there are many problems where the structure of the Jacobian matrix is known in advance. As we showed in Section 4.1, the group word problem can always be simulated with permutation matrices for its dynamics. As we showed in Section 4.3, discretized roll-outs from differential equations, used in sampling from diffusion models and rolling out neural ODEs, have $\partial f/\partial x$ equal to the identity matrix plus a correction term proportional to the discretization step-size. Moreover, as shown in Zoltowski et al. (2025), the dynamics of position and momenta variables in Hamiltonian Monte Carlo (HMC) results in banded matrices. Furthermore, in sequence modeling, one can *design* a recurrent neural network to have Jacobians with desired structure (Farsang & Grosu, 2025; Zattra et al., 2025; Danieli et al., 2026). Finally, if there is truly no analytic information about the Jacobian in advance, its structure can be probed with finite-difference methods.

**Future directions** Clarifying the relationships and properties of these approaches through the lens of linear dynamical systems also suggests promising areas for future study. One clear direction of future work is to explore additional approaches for exploiting problem-specific structure, using our unifying framework to develop new fixed-point iterations. For example, an intermediate between Picard and quasi-Newton methods is a scaled identity approximation, $\tilde{A}_{t+1} = a_{t+1}I_D$. If we had prior knowledge on the appropriate scaling factors, $a_{t+1} \in \mathbb{R}$, we could avoid computing any Jacobian-vector product evaluations. More generally, there exist other groups of structured matrices with compact representations that are closed under composition such that a parallel evaluation of the LDS would be computationally efficient. Examples include permutation matrices, block-diagonal matrices, and block matrices where each sub-block is diagonal, among others. Future work should enumerate these use cases and investigate problem-specific applications where they are appropriate. One example application is for more efficient parallelization of the group word problem using a compact representation of permutation matrices, as was done in concurrent work by Terzić et al. (2025).

Understanding the shared backbone of these fixed-point methods can give practitioners guidance about which methods to use for which problems. As parallel evaluation of seemingly sequential processes becomes increasingly important in machine learning, these insights may provide valuable guidance to the field.

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

## A    Author Contributions and Acknowledgments

**Author Contributions**   XG, EKB, and SWL conceived the project. XG and SWL proved Proposition 1 and Proposition 2. XG provided most of the content of Section 3, with input from all authors. XG and JWL proved Proposition 4. HDL developed, ran, and wrote up the state tracking experiments. XG, EKB, HDL, JWL, and KAW all wrote substantial code in various exploratory aspects of the project. DZ and CR contributed important insights about the relationships between algorithms and computational efficiency. LK contributed important insights about rates of convergence and the importance of stability. All authors made important contributions to conceptualization, brainstorming, exploration, writing, and editing. XG, EKB, and SWL wrote the manuscript with input from all authors. SWL supervised the project.

**Acknowledgments**   We thank Nicolas Zucchet, Francois Chaubard, Libby Zhang, Henry Smith, and Ian Christopher Tanoh, as well as the anonymous reviewers on Open Review, for helpful feedback. X.G. acknowledges support from the Walter Byers Graduate Scholarship from the NCAA. J.W.L. was supported by the Department of Energy Computational Science Graduate Fellowship under Award Number DE-SC0023112. S.W.L. was supported by fellowships from the Simons Collaboration on the Global Brain, the Alfred P. Sloan Foundation, and the McKnight Foundation. We thank Stanford University and the Stanford Research Computing Center for providing computational resources and support. Additional computations were performed on Marlowe (Kapfer et al., 2025), Stanford University's GPU-based Computational Instrument, supported by Stanford Data Science and Stanford Research Computing. The authors have no competing interests.

## B    The Parallel Scan: A Gentle Introduction

### B.1    A very simple example: multiplying matrices

The *parallel scan* (Stone, 1973; Blelloch, 1990), also known as the *associative* scan and, colloquially, *pscan*, is a well-known primitive in the parallel computing literature (Hillis & Steele Jr, 1986; Ladner & Fischer, 1980; Lakshmivarahan & Dhall, 1994). The core idea of the parallel scan is a divide-and-conquer algorithm. We illustrate this point in the simple example of multiplying a series of matrices together.

**Simple Example (Multiplying Matrices):**   Given a series of square matrices $A_1, A_2, \ldots, A_{T-1}, A_T$, compute their product[5], $A_T A_{T-1} \ldots A_2 A_1$. The simplest way to carry out the matrix multiplication is sequentially: first compute $A_1$, then compute $A_2 A_1$, then compute $A_3 A_2 A_1$, and so on. Such an approach takes $\mathcal{O}(T)$ time.

A core insight of the parallel scan is that matrix multiplication is *closed*; that is, if $A_s \in \mathbb{R}^{D \times D}$ and $A_t \in \mathbb{R}^{D \times D}$, then $A_t A_s \in \mathbb{R}^{D \times D}$. Thus, matrix products can be computed recursively in pairs, as illustrated in Figure 5.

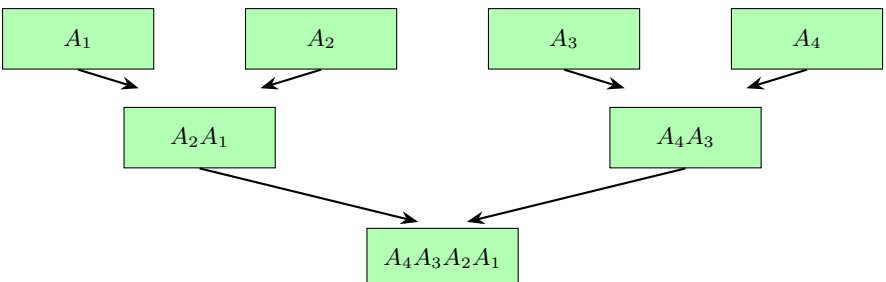

Figure 5: **Parallel Scan for Matrix Multiplication.** We illustrate a divide-and-conquer approach to compute the product $A_4 A_3 A_2 A_1$. Note that this divide-and-conquer approach naturally leads to $\mathcal{O}(\log T)$ depth.

---

[5]Note that we have the matrices act via left-multiplication over the sequence length, because this is the most common way to write matrix-vector products.

Because of the divide-and-conquer (binary-tree-like) nature of this approach to multiplying matrices, with $\mathcal{O}(T)$ processors, the time needed to get the matrix product is only $\mathcal{O}(\log T)$. This simple example illustrates the core intuition behind the parallel scan: a closed operation leading to a divide-and-conquer approach that parallelizes a computation so that it takes sublinear time. However, there are two additional details of the parallel associative scan that we should address: arbitrary binary associative operators and closure; and getting intermediate products.

### B.2   Detail #1: Parallel scans for arbitrary binary associative operators

Matrix multiplication is an associative operator, as $A_3(A_2A_1) = (A_3A_2)A_1$. In general, consider a binary associative operator $\otimes$, which would satisfy $q_3 \otimes (q_2 \otimes q_1) = (q_3 \otimes q_2) \otimes q_1$. Now, let us further assume that this binary associative operator is closed:

**Definition 2** (Closure). *A binary associative operator $\otimes$ is closed over a set $\mathcal{S}$ if it satisfies the property:*

$$q_1 \in \mathcal{S}, q_2 \in \mathcal{S} \Rightarrow q_2 \otimes q_1 \in \mathcal{S}. \tag{18}$$

If $\otimes$ is closed, then we can again use a parallel scan to compute the cumulative product of the operands. A wide range of binary associative operators are closed, and can thus be parallelized with the parallel scan. Some examples include:

- **Scalar addition:** The fact that addition of scalars (and vectors) is closed allows cumulative sums to be computed with the parallel scan algorithm. In this instance, it is also known as the *prefix sum* algorithm. Clearly, addition is associative and closed, and so summing a series of scalars can be done with a divide-and-conquer approach.

- **Composition of affine functions, as in LDSs.** Consider the affine function $f_i(x) = A_i x + b_i$. Notice that the composition of affine functions is also affine, as $f_j(f_i(x)) = A_j A_i x + (b_j + A_j b_i)$. Thus, if we represent the operands as ordered pairs $(A_i, b_i)$ and $(A_j, b_j)$, we can write the associative operator $\otimes$ for the composition of affine functions as

  $$(A_i, b_i) \otimes (A_j, b_j) = (A_j A_i, b_j + A_j b_i).$$

  Thus, we observe that in this setting, $\otimes$ is closed. We also should check that $\otimes$ is associative: we can do so with either elementary algebra, or by observing that function composition is associative.

  This observation that composition of affine functions can be parallelized with the associative scan is what lets us parallelize LDSs. The parallelization of LDSs is what allows for parallel computation in many important architectures in sequence modeling, such as linear RNNs (Martin & Cundy, 2018; Orvieto et al., 2023), deep SSMs (Smith et al., 2023; Gu & Dao, 2024), and nonlinear[6] RNNs (Lim et al., 2024; Gonzalez et al., 2024; Farsang & Grosu, 2025; Danieli et al., 2026). This parallel scan for LDSs is the core primitive uniting the fixed-point methods discussed in this paper.

- **Kalman filtering and smoothing:** Parallel scans can also be utilized in probabilistic modeling. A standard probabilistic model is the *linear Gaussian state space model* (LGSSM), where the latent variables $x_t$ follow linear dynamics with Gaussian noise, and emit observations $y_t$ with linear readouts with Gaussian noise (Murphy, 2023; Särkkä & Svensson, 2023). See Figure 6.

  Two canonical inferential targets in the LGSSM are the filtering distributions, $p(x_t \mid y_{1:t})$, and the smoothing distributions, $p(x_t \mid y_{1:T})$. The Kalman filter (Kalman, 1960) and Rauch-Tung-Striebel (RTS) smoother (Rauch et al., 1965) obtain the filtering and smoothing distributions (respectively) in an LGSSM. The Kalman filter makes a single pass forward in time to get the filtering distributions, while the RTS smoother then makes an additional pass backwards in time to get the smoothing distributions. Both the Kalman filter and RTS smoother would seem to be inherently sequential algorithms, requiring $\mathcal{O}(T)$ time. However, Särkkä & García-Fernández (2021) demonstrated that

---

[6] The parallel scan is used in nonlinear RNNs via the iterative fixed-point methods discussed in this paper, i.e. Newton and quasi-Newton iterations, see Algorithm 1.

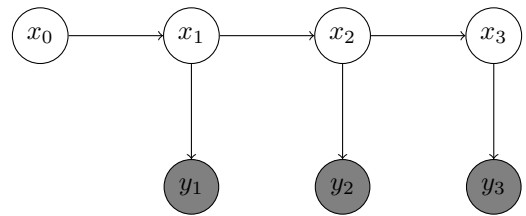

Figure 6: **A linear Gaussian state space model (LGSSM):** The LGSSM consists of latent variables $x_t$ and observed variables $y_t$. The generative model of the LGSSM consists of dynamics $x_{t+1} \sim \mathcal{N}(Ax_t, Q)$ and emissions $y_{t+1} \sim \mathcal{N}(Hx_{t+1}, R)$.

the Kalman filter and RTS smoother can also be parallelized over the sequence length via the construction of custom binary associative operators and a parallel scan. While we leave the details of this construction to Särkkä & García-Fernández (2021), we note that it is intuitively plausible to be able to parallelize filtering and smoothing in an LGSSM with a parallel scan because

- the dynamical backbone is an LDS, for which we have a parallel scan;
- since everything is linear and Gaussian, all distributions remain Gaussian, hinting at closure; and
- we can combine $p(x_s|x_0, y_{1:s})$ with $p(x_t|x_s, y_{s+1:t})$ to obtain $p(x_t|x_0, y_{1:t})$, suggesting a divide-and-conquer strategy.

These parallel filtering and smoothing algorithms are useful in machine learning, allowing for parallelization of structured variational autoencoders (Johnson et al., 2016; Zhao & Linderman, 2023). Similar approaches also work for Hidden Markov Models (Hassan et al., 2021) and for computing log-normalizing constants (Hu et al., 2025).

- **Parallelizing fixed point iterations:** Finally, these parallel filtering and smoothing are directly applicable to the types of parallel fixed-point iterations that are the focus of this paper. In particular, the ELK algorithm (Gonzalez et al., 2024)—**E**valuating **L**evenberg-Marquardt via **K**alman— stabilizes the Newton iterations (Lim et al., 2024) we discuss in this paper using the Levenberg-Marquardt (trust-region) method (Levenberg, 1944; Marquardt, 1963). The trust-region updates are able to be computed using a parallel scan because they are equivalent to a Kalman smoother in an appropriately constructed LGSSM (Särkkä & Svensson, 2020).

**What about arbitrary function composition?** The astute reader might note that the composition of functions, i.e. $f_1 \circ f_2$, is *always* a binary associative operator. So, why do we have all these special cases of parallel scans, and not simply one parallel scan for the composition $\circ$ of arbitrary functions $f_i$?

The reason to have many different parallel scans is precisely the importance of having the binary associative operator be *closed*. In all the previous examples, the binary associative operator $\otimes$ satisfies Definition 2, letting us easily store combinations of operands $q_i \otimes q_j$ and so employ a divide-and-conquer technique.

While we could consider some gigantic function space $\mathcal{F}$, for which function composition would be closed, the practical question then becomes: how would we store the combinations of operands? If we do not have some compact representation for elements of $\mathcal{F}$, then we cannot use a parallel scan in practice, even though the parallel scan may seem applicable in theory.

### B.3 Detail #2: Obtaining the intermediate terms in the product

When we evaluate a linear dynamical system, we often do not want only the final term $x_T$, but also all the intermediate terms $\mathbf{x}_{1:T}$, i.e. the full trajectory. So far, the parallel scan as presented would only yield the final term $x_T$ as well as intermediate terms that happened to be powers of 2, i.e. $x_1, x_2, x_4, x_8$, etc.

However, the parallel scan can be easily adjusted to obtain all the intermediate terms as well. Let us return to our very simple example of matrix multiplication to illustrate, in particular the setting where $T = 8$. We will denote the individual matrices as $A_1, A_2, A_3, \ldots A_8$, and their products as $A_{s:t}$, i.e. $A_{5:6} = A_6 A_5$.

The first phase of the parallel scan is the *up-sweep*, and takes $\log(T)$ iterations and $\mathcal{O}(T)$ memory. We start multiplying adjacent pairs of matrices together, going, for example[7], from $A_8$ to $A_{7:8}$ to $A_{5:8}$ to $A_{1:8}$.

Then, in the *down-sweep*, we fill in the missing products to obtain all the cumulative products $A_{1:t}$ for $1 \leq t \leq T$. Intuitively, the down-sweep also takes $\mathcal{O}(\log T)$ iterations, for the same reason that any natural number $T$ can be represented using $1 + \log_2(T)$ digits in binary.

Table 2: **Up-sweep** for multiplying $A_1, A_2, \ldots A_8$.

| Position 1 | Position 2 | Position 3 | Position 4 | Position 5 | Position 6 | Position 7 | Position 8 |
|---|---|---|---|---|---|---|---|
| $A_1$ | $A_2$ | $A_3$ | $A_4$ | $A_5$ | $A_6$ | $A_7$ | $A_8$ |
| $A_1$ | $A_{1:2}$ | $A_3$ | $A_{3:4}$ | $A_5$ | $A_{5:6}$ | $A_7$ | $A_{7:8}$ |
| $A_1$ | $A_{1:2}$ | $A_3$ | $A_{1:4}$ | $A_5$ | $A_{5:6}$ | $A_7$ | $A_{5:8}$ |
| $A_1$ | $A_{1:2}$ | $A_3$ | $A_{1:4}$ | $A_5$ | $A_{5:6}$ | $A_7$ | $A_{1:8}$ |

Table 3: **Down-sweep** for multiplying $A_1, A_2, \ldots A_8$.

| Position 1 | Position 2 | Position 3 | Position 4 | Position 5 | Position 6 | Position 7 | Position 8 |
|---|---|---|---|---|---|---|---|
| $A_1$ | $A_{1:2}$ | $A_3$ | $A_{1:4}$ | $A_5$ | $A_{1:6}$ | $A_7$ | $A_{1:8}$ |
| $A_1$ | $A_{1:2}$ | $A_{1:3}$ | $A_{1:4}$ | $A_{1:5}$ | $A_{1:6}$ | $A_{1:7}$ | $A_{1:8}$ |

Thus, together, the up-sweep and the down-sweep of the parallel scan run in $\mathcal{O}(\log T)$ time on $\mathcal{O}(T)$ processors, and at the end of this algorithm, we get all of the intermediate products[8] (the "prefix sums").

### B.4 Implementation considerations

This gentle introduction provides the main ideas and intuition of the parallel scan: closed binary associative operators leading a divide-and-conquer algorithm to leverage parallel processors to compute sequential compositions in sublinear time. However, there are also a host of implementation details for using the parallel scan when programming on accelerated hardware like GPUs (Harris et al., 2007; Yang et al., 2024; Sarnthein, 2025). For example, the presence of a general-purpose parallel scan is, as of the time of writing, a major difference between JAX (Bradbury et al., 2018) and PyTorch (Paszke et al., 2019), two leading Python libraries for deep learning. JAX has a general purpose parallel scan (`jax.lax.associative_scan`) as a fundamental primitive, which allows for implementation of a wide range of parallel scans. For example, `dynamax`, a JAX library for probabilistic state space modeling (Linderman et al., 2025), implements the parallel filtering and smoothing algorithms from Särkkä & García-Fernández (2021). In contrast, PyTorch currently has only `torch.cumsum`, which is the parallel scan where the binary associative operator is addition[9], and `torch.cumprod` (for scalar multiplication). This difference is why we implement the experiments in this paper in JAX. This lack of a general purpose parallel scan in PyTorch has also led to the custom development of highly-optimized, hardware-aware custom CUDA kernels for parallel scans. These custom parallel scans appear most prominently in Mamba (Gu & Dao, 2024), a leading SSM for language modeling, and ParaRNN (Danieli et al., 2026), which applies the Newton iterations discussed in this paper to 7B parameter nonlinear RNNs to achieve strong language modeling performance. There also exist useful implementations of parallel scans for scalar/diagonal LDSs in PyTorch such as Kyrylov (2024), which we used to implement quasi-Newton iterations in PyTorch in this repo: `https://github.com/lindermanlab/elk-torch`

---

[7]See Position 8 of Table 2

[8]See the last row of Table 3.

[9]Although Heinsen (2023) shows that clever uses of `torch.cumsum` can parallelize scalar/diagonal LDSs, of the type that are used in quasi-Newton iterations (eq. (6)).

## C  Further Discussion of Convergence Analysis

In this appendix, we provide a deeper discussion of the details, intuitions, and limitations of the convergence analysis presented in Section 3.

### C.1  Formula for approximate Jacobian

For completeness, we provide the formula for the block-bidiagonal approximate Jacobian $\widetilde{\mathbf{J}}(\mathbf{x}_{1:T}) \in \mathbb{R}^{TD \times TD}$ that the fixed-point methods considered in this paper give rise to.

$$
\widetilde{\mathbf{J}}(\mathbf{x}_{1:T}) := \begin{pmatrix}
I_D & 0 & 0 & \dots & 0 & 0 \\
-\tilde{A}_2(x_1) & I_D & 0 & \dots & 0 & 0 \\
0 & -\tilde{A}_3(x_2) & I_D & \dots & 0 & 0 \\
\vdots & \vdots & \vdots & \ddots & \vdots & \vdots \\
0 & 0 & 0 & \dots & I_D & 0 \\
0 & 0 & 0 & \dots & -\tilde{A}_T(x_{T-1}) & I_D
\end{pmatrix}. \tag{19}
$$

For Newton's method $\mathcal{A}_N$, the resulting $\widetilde{\mathbf{J}}_N(\mathbf{x}_{1:T}) = \mathbf{J}(\mathbf{x}_{1:T})$, where $\mathbf{J}(\mathbf{x}_{1:T})$ is defined in eq. (11). For Picard iterations, $\widetilde{\mathbf{J}}_P(\mathbf{x}_{1:T})$ takes the form

$$
\widetilde{\mathbf{J}}_P(\mathbf{x}_{1:T}) = \begin{pmatrix}
I_D & 0 & 0 & \dots & 0 & 0 \\
-I_D & I_D & 0 & \dots & 0 & 0 \\
0 & -I_D & I_D & \dots & 0 & 0 \\
\vdots & \vdots & \vdots & \ddots & \vdots & \vdots \\
0 & 0 & 0 & \dots & I_D & 0 \\
0 & 0 & 0 & \dots & -I_D & I_D
\end{pmatrix}. \tag{20}
$$

For Jacobi iterations, $\widetilde{\mathbf{J}}_J(\mathbf{x}_{1:T})$ is always the identity matrix $I_{TD}$.

### C.2  Limitations of Proposition 3

Proposition 3 only guarantees a decrease in the error when the iterate $\mathbf{x}_{1:T}^{(i)}$ is already in a basin of decrease $\mathcal{B}_D$ given by

$$
\mathcal{B}_D := \left\{ \mathbf{x}_{1:T} : \|\mathbf{e}(\mathbf{x}_{1:T})\|_2 \le 2 \cdot \frac{1 - \left\| \widetilde{\mathbf{J}}(\mathbf{x}_{1:T})^{-1} \right\|_2 \left\| \widetilde{\mathbf{J}}(\mathbf{x}_{1:T}) - \mathbf{J}(\mathbf{x}_{1:T}) \right\|_2}{L \left\| \widetilde{\mathbf{J}}(\mathbf{x}_{1:T})^{-1} \right\|_2} \right\}.
$$

However, since we know from Proposition 1 of Gonzalez et al. (2024) that all the fixed-point algorithms considered in this paper must eventually converge, we know that the iterates $\mathbf{x}_{1:T}^{(i)}$ must all eventually enter this basin of decrease $\mathcal{B}_D$ if $\mathcal{B}_D \ne \emptyset$. For this reason, Proposition 3 provides helpful intuition about which fixed-point algorithms are useful for which dynamical systems.

For example, let us define the basin of linear rate $\mathcal{B}_L$ to comprise those $\mathbf{x}_{1:T}$ where $\left\| \widetilde{\mathbf{J}}(\mathbf{x}_{1:T}) - \mathbf{J}(\mathbf{x}_{1:T}) \right\|_2 \|\mathbf{e}_{1:T}^{(i)}\|_2 > \frac{L}{2}(\|\mathbf{e}_{1:T}^{(i)}\|_2^2)$, i.e. the expression linear in $\|\mathbf{e}_{1:T}^{(i)}\|_2$ on right side of equation 13 dominates the expression quadratic in $\|\mathbf{e}_{1:T}^{(i)}\|_2$. It follows that $\mathcal{B}_L$ is given by

$$
\mathcal{B}_L := \left\{ \mathbf{x}_{1:T} : \|\mathbf{e}(\mathbf{x}_{1:T})\|_2 \le \frac{2 \left\| \widetilde{\mathbf{J}}(\mathbf{x}_{1:T}) - \mathbf{J}(\mathbf{x}_{1:T}) \right\|_2}{L} \right\}.
$$

Therefore, when $\mathbf{x}_{1:T}^{(i)} \in \mathcal{B}_D \cap \mathcal{B}_L$, it follows that the norm of the error is guaranteed to decrease by a factor of $2 \left\| \widetilde{\mathbf{J}}(\mathbf{x}_{1:T}^{(i)})^{-1} \right\|_2 \left\| \widetilde{\mathbf{J}}(\mathbf{x}_{1:T}^{(i)}) - \mathbf{J}(\mathbf{x}_{1:T}^{(i)}) \right\|_2$. Moreover, as $\|\mathbf{e}_{1:T}^{(i)}\|_2$ approaches zero, the guaranteed factor of decrease approaches the value given by eq. (15).

### C.3 Extended discussion about intuitions from the rate of convergence

In this appendix, we elaborate on the intuitions about the rate of convergence presented in Section 3.3.

#### C.3.1 Intuitions from Jacobian approximation error

First, elaborating on the section discussing $\left\|\widetilde{\mathbf{J}}(\mathbf{x}_{1:T}) - \mathbf{J}(\mathbf{x}_{1:T})\right\|_2$, we provide a proof of Lemma 1.

*Proof of Lemma 1.* Plugging in the functional forms of $\widetilde{\mathbf{J}}(\cdot)$ and $\mathbf{J}(\cdot)$, if we define $E_{t+1} := \tilde{A}_{t+1}(x_t) - A_{t+1}(x_t)$, then

$$\widetilde{\mathbf{J}}(\mathbf{x}_{1:T}) - \mathbf{J}(\mathbf{x}_{1:T}) = \begin{pmatrix} 0 & 0 & 0 & \dots & 0 & 0 \\ E_2 & 0 & 0 & \dots & 0 & 0 \\ 0 & E_3 & 0 & \dots & 0 & 0 \\ \vdots & \vdots & \vdots & \ddots & \vdots & \vdots \\ 0 & 0 & 0 & \dots & 0 & 0 \\ 0 & 0 & 0 & \dots & E_T & 0 \end{pmatrix}.$$

The spectral norm of a matrix $M$ is equal to the square root of the largest eigenvalue of $M^\top M$. Defining $M := \widetilde{\mathbf{J}}(\mathbf{x}_{1:T}) - \mathbf{J}(\mathbf{x}_{1:T})$, we see that

$$M^\top M = \begin{pmatrix} 0 & 0 & 0 & \dots & 0 & 0 \\ 0 & E_2^\top E_2 & 0 & \dots & 0 & 0 \\ 0 & 0 & E_3^\top E_3 & \dots & 0 & 0 \\ \vdots & \vdots & \vdots & \ddots & \vdots & \vdots \\ 0 & 0 & 0 & \dots & E_{T-1}^\top E_{T-1} & 0 \\ 0 & 0 & 0 & \dots & 0 & E_T^\top E_T \end{pmatrix}.$$

Since $M^\top M$ is a block-diagonal matrix, its eigenvalues are equal to the union of the eigenvalues of each of the blocks $E_t^\top E_t$. Thus, it follows that the maximum eigenvalue of $M^\top M$ is equal to the maximum of all the eigenvalues of all the matrices $E_t^\top E_t$, and so the maximum singular value of $\widetilde{\mathbf{J}}(\mathbf{x}_{1:T}) - \mathbf{J}(\mathbf{x}_{1:T})$ is given by $\max_{1 \le t \le T-1} \left\|\tilde{A}_{t+1}(x_t) - A_{t+1}(x_t)\right\|_2$. $\qquad\square$

#### C.3.2 Intuitions from the norm of the inverse of the approximate Jacobian

Next, we elaborate on our bounds for $\left\|\widetilde{\mathbf{J}}(\mathbf{x}_{1:T})^{-1}\right\|_2$.

Recall that $\widetilde{\mathbf{J}}(\mathbf{x}_{1:T})^{-1}$ takes the form (Dao & Gu, 2024; Gonzalez et al., 2025)

$$\widetilde{\mathbf{J}}(\mathbf{x}_{1:4})^{-1} = \begin{pmatrix} I_D & 0 & 0 & 0 \\ \tilde{A}_2 & I_D & 0 & 0 \\ \tilde{A}_3 \tilde{A}_2 & \tilde{A}_3 & I_D & 0 \\ \tilde{A}_4 \tilde{A}_3 \tilde{A}_2 & \tilde{A}_4 \tilde{A}_3 & \tilde{A}_4 & I_D \end{pmatrix},$$

shown above for $T = 4$.

Theorem 2 of Gonzalez et al. (2025) controls $\left\|\widetilde{\mathbf{J}}(\mathbf{x}_{1:T})^{-1}\right\|_2$ in terms of $\overline{\rho} := \sup_{2 \le t \le T, x \in \mathbb{R}^D} \|\tilde{A}_t(x)\|_2$ and $\underline{\rho} := \inf_{2 \le t \le T, x \in \mathbb{R}^D} \|\tilde{A}_t(x)\|_2$ as

$$\max(1, \underline{\rho}^{T-1}) \le \|\widetilde{\mathbf{J}}(\mathbf{x}_{1:T})^{-1}\|_2 \le \frac{1 - \overline{\rho}^T}{1 - \overline{\rho}},$$

for $\overline{\rho} \ne 1$. We would therefore expect that fixed-point methods that give rise to unstable LDSs with transition matrices having spectral norms much larger than one should have slower rates of convergence. Methods that give rise to unstable LDSs suffer from numerical blowup, especially for large $T$.

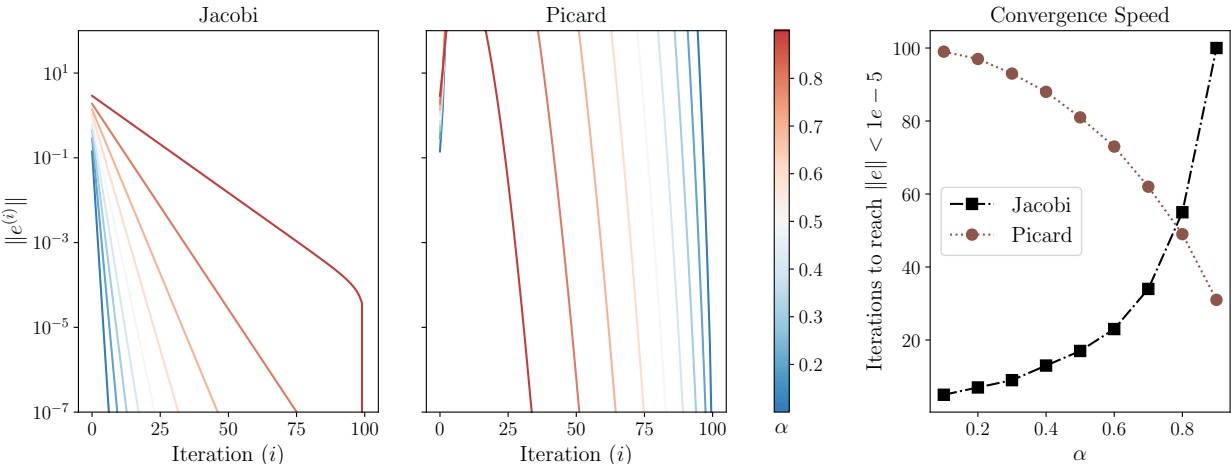

Figure 7: **Comparing Picard and Jacobi iterations on a diagonal LDS.** For the underlying dynamical system $x_{t+1} = \alpha x_t$ with $T = 100$, we plot the norm of the error $\mathbf{e}_{1:T}^{(i)}$ for Jacobi **(Left)** and Picard iterations **(Center)**. In the **(Right)** panel, we also show the number of iterations needed for the norm of the error to go below $1 \times 10^{-5}$.

Moreover, in the special cases of Jacobi and Picard iterations, we can compute $\left\|\widetilde{\mathbf{J}}(\mathbf{x}_{1:T}^{(i)})^{-1}\right\|_2$ analytically. For Jacobi iterations, $\left\|\widetilde{\mathbf{J}}_J\right\|_2 = 1$. For Picard iterations, the expression for $\left\|\widetilde{\mathbf{J}}_P\right\|_2$ is more complicated, but it scales as $O(T)$ (see Lemma 2 in Appendix D.1).

Because $\|\widetilde{\mathbf{J}}_P^{-1}\|_2 > \|\widetilde{\mathbf{J}}_J^{-1}\|$ for large $T$, the formula for $\gamma$ given by eq. (15) yields the following expectation:

> In settings where the $A_{t+1}$ from Picard vs. Jacobi iterations approximates the true dynamics Jacobian $\partial f_{t+1}/\partial x_t$ equally well, we expect Jacobi iterations to converge more quickly because $\|\widetilde{\mathbf{J}}_J^{-1}\| < \|\widetilde{\mathbf{J}}_P^{-1}\|_2$.

We test this hypothesis in the next section with a simple simulation designed to show how Proposition 3 provides helpful intuition about the convergence rates of different fixed-point methods.

### C.4   A simulation using Proposition 3 to distinguish between Jacobi and Picard iterations

We demonstrate the helpfulness of the intuitions stemming from Proposition 3 in a simple simulation. We consider the LDS $x_{t+1} = \alpha x_t$, for $x_t \in \mathbb{R}^2$. Because this is an LDS with diagonal dynamics, both the Newton and quasi-Newton iterations considered in this paper converge in one iteration. However, this simulation is useful for comparing Jacobi versus Picard iterations. This comparison is particularly fruitful in light of the formula for $\gamma$ given by eq. (15) and Lemma 1 because, in this setting,

$$\|\widetilde{\mathbf{J}}_J - \mathbf{J}\|_2 = \alpha$$
$$\|\widetilde{\mathbf{J}}_P - \mathbf{J}\|_2 = 1 - \alpha.$$

However, $\|\widetilde{\mathbf{J}}_J^{-1}\|_2 = 1$, while $\|\widetilde{\mathbf{J}}_P^{-1}\|_2$ scales linearly with $T$. Therefore, when comparing the number of Jacobi iterations needed to converge when the dynamics are multiplication by $\alpha$ to the number of Picard iterations needed to converge when the dynamics are multiplication by $1 - \alpha$, we expect fewer Jacobi iterations should be needed than Picard iterations, as $\gamma_J < \gamma_P$.

We observe precisely this behavior in Figure 7. For $\alpha = 0.5$, when $\|\widetilde{\mathbf{J}}_J - \mathbf{J}\|_2 = \|\widetilde{\mathbf{J}}_P - \mathbf{J}\|_2$ , we see that Jacobi iterations converge in far fewer iterations than Picard iterations. Moreover, when comparing the behavior of

Jacobi for simulating $f_{t+1}(x_t) = \alpha x_t$ with Picard for simulating $f_{t+1}(x_t) = (1-\alpha)x_t$, we observe that Jacobi iterations always converge faster. However, when comparing for the same value of $\alpha$, we see that Picard can be faster than Jacobi when $\alpha$ is closer to one. This behavior makes sense, because in those settings the true Jacobian $\partial f_{t+1}/\partial x_t$ is closer to $I_D$ than to $\mathbf{0}$.

Moreover, we observe that in this setting, the error $\mathbf{e}_{1:T}^{(i)}$ for Jacobi iteration shows a clear linear convergence rate, as predicted by Proposition 3. The slope of the norm of the errors of the Jacobi iterates should be $\log_{10}(\alpha)$ by eq. (15) and Lemma 1, and in fact those values are exactly the slopes of the lines in Figure 7 (Left panel).

## D   Theoretical Details: Additional Proofs

### D.1   Lemma 2

**Lemma 2.** *Let $\widetilde{\mathbf{J}}_P$ be as in eq. (20). Then*

$$\|\widetilde{\mathbf{J}}_P^{-1}\|_2 = \frac{1}{2\sin\left(\frac{\pi}{2(2T+1)}\right)}$$

*By the small-angle approximation for sine, $\|\widetilde{\mathbf{J}}_P^{-1}\|_2$ scales as $\mathcal{O}(T)$.*

*Proof.* Consider

$$K := \widetilde{\mathbf{J}}_P^{-\top}\widetilde{\mathbf{J}}_P^{-1} = \begin{pmatrix} I_D & I_D & I_D & \dots & I_D \\ I_D & 2I_D & 2I_D & \dots & 2I_D \\ I_D & 2I_D & 3I_D & \dots & 3I_D \\ \vdots & \vdots & \vdots & \ddots & \vdots \\ I_D & 2I_D & 3I_D & \dots & TI_D \end{pmatrix}.$$

We know that $\lambda_{\max}(K)^{1/2} = \|\widetilde{\mathbf{J}}_P^{-1}\|_2$. Since $K$ is a Kronecker product $M \otimes I_D$, where $M_{i,j} = \min(i,j)$, the spectrum of $K$ is equivalent to the spectrum of $M$ (just with all eigenvalues having multiplicity $D$). Therefore, we seek to find the spectrum of $M \in \mathbb{R}^{T \times T}$.

However, the spectrum of $M$ is known in the literature. For example, Theorem 2.1 of da Fonseca (2007) shows that if $T \geq 3$, then the eigenvalues $\{\lambda_k\}_{k=0}^{T-1}$ of $M$ are given by

$$\lambda_k = \frac{1}{2}\left(1 - \cos\left(\frac{2k+1}{2T+1}\pi\right)\right)^{-1}$$

$$= \frac{1}{4}\left(\sin\left(\frac{2k+1}{2(2T+1)}\pi\right)\right)^{-2}$$

where the second equality comes from the half-angle formula. We observe that the largest eigenvalue is therefore $\lambda_0$, and so the result follows after we take a square root. $\qquad\square$

## E   Experimental Details and Additional Experiments

We implemented our experiments using the Equinox library (Kidger & Garcia, 2021) in JAX (Bradbury et al., 2018). In our experiments that report wall-clock time, we use a stochastic implementation of quasi-Newton iterations (Zoltowski et al., 2025) that estimates the diagonal using the Hutchinson estimator (Hutchinson, 1989); see Section 3.4 of Zoltowski et al. (2025) for details.

The stopping criterion we use for deciding when the fixed-point iterations in eq. (2) have converged is based on the merit function $\mathcal{L}(\mathbf{x}_{1:T})$, which is defined as:

$$\mathbf{r}(\mathbf{x_{1:T}}) = [x_1 - f(x_0), x_2 - f(x_1), x_3 - f(x_2), \dots, x_T - f(x_{T-1})]$$

$$\mathcal{L}(\mathbf{x_{1:T}}) = \frac{1}{2}\|\mathbf{r}(\mathbf{x_{1:T}})\|_2^2.$$

In our experiments we use a tolerance of $5 \times 10^{-4}$, that is we terminate the fixed-point iterations when iterate $i$ satisfies

$$\mathcal{L}(\mathbf{x}_{1:T}^{(i)}) \leq 5 \times 10^{-4}.$$

### E.1 Experimental Details for Appendix C.4

For each scalar LDS with scalar multiplication $\alpha$, we use a state size of $D = 2$, i.e. $x_t \in \mathbb{R}^2$, and a sequence length of $T = 100$. We consider 10 random seeds, where the randomness controls the initial state $x_0$. We initialize $\mathbf{x}_{1:T}^{(0)}$ at all zeros. We plot the median number of steps to convergence in Figure 7.

### E.2 Experimental Details for Section 4.1

We use 10 random seeds, where the randomness controls the sequence of the $S_5$ group elements. Each seed uses a batch size of 16, i.e. 16 different $S_5$ word problems are evaluated for each run. For each seed, we time how long it takes for 5 runs of the fixed point solver (sequential, Picard, quasi-Newton, or Newton) to evaluate, and record this mean wall-clock time. In Figure 2, we then plot the median of these 10 mean wall-clock times. We run on an H100 with 80GB of onboard memory.

### E.3 Experimental Details for Section 4.2

The experiment depicted in Figure 3 closely follows the experiments in Section 4.1 of Lim et al. (2024) and Section 6.1 of Gonzalez et al. (2024, Section 6.1). We use 10 random seeds, where the randomness controls the initialization of the GRUs, the random inputs to the GRUs, and Rademacher variables used in the stochastic implementation of quasi-Newton iterations (Zoltowski et al., 2025). The GRUs were initialized following the standard initialization practice in Equinox. Each seed uses a batch size of 16, i.e. 16 different GRU trajectories are evaluated for each run. For each seed, we time how long it takes for 5 runs of the fixed point solver (sequential, Picard, quasi-Newton, or Newton) to evaluate, and record this mean wall-clock time. In Figure 3, we then plot the median of these 10 mean wall-clock times. We run on an H100 with 80GB of onboard memory.

To demonstrate the different values of the $\mathrm{diff}(\cdot)$ operator for quasi-Newton, Jacobi, and Picard iterations in this setting, we consider the setting $D = 8$ and $T = 1000$. In Figure 8, we plot a variety of quantities relevant for the $\gamma$ (cf. eq. (15)) for ten random seeds. We observe that lower values of $\gamma$ (i.e., faster rates of asymptotic linear convergence) coincide with fewer fixed-point iterations needed in Figure 3. We observe that both $\mathrm{diff}(\mathcal{A}_J)$ and $\mathrm{diff}(\mathcal{A}_{QN})$ are both below one always, which corresponds to their fast rates of convergence demonstrated in Figure 3. In contrast, $\mathrm{diff}(\mathcal{A}_P)$ is always greater than one, which corresponds to the slow rates of convergence of Picard iteration in the experiment depicted in Figure 3.

### E.4 Experimental Details for Section 4.3

The potential $\phi$ used for the experiment depicted in Figure 4 is the negative log probability of a mixture of Gaussians. Each Gaussian is $D$-dimensional, and has a random covariance matrix drawn from a Wishart distribution. For each fixed-point method, we ran 10 random seeds, where the randomness controls the randomly chosen covariance matrices for the mixture of Gaussians and the random inputs for Langevin dynamics. Each seed uses a batch size of 16, i.e. 16 different Langevin trajectories are evaluated for each run. We use a step size $\epsilon = 1 \times 10^{-5}$ for the discrete Langevin steps. For each seed, we time how long it takes for 5 runs of the fixed point solver (sequential, Picard, quasi-Newton, or Newton) to evaluate, and record this mean wall-clock time. In Figure 4, we then plot the median of these 10 mean wall-clock times. We run on an H100 with 80GB of onboard memory. In order to get convergence for Newton iterations for longer sequence lengths, we had to run with increased precision, using the `highest` option for the `jax_default_matmul_precision` flag. See Appendix E.5 for further discussion of the importance of numerical precision for implementation of these LDS-based fixed-point methods.

We also include a small additional experiment in Figure 9: instead of varying the state size dimension, we instead vary $K$, the number of Gaussians that make up the potential $\phi$ determining the Langevin dynamics.

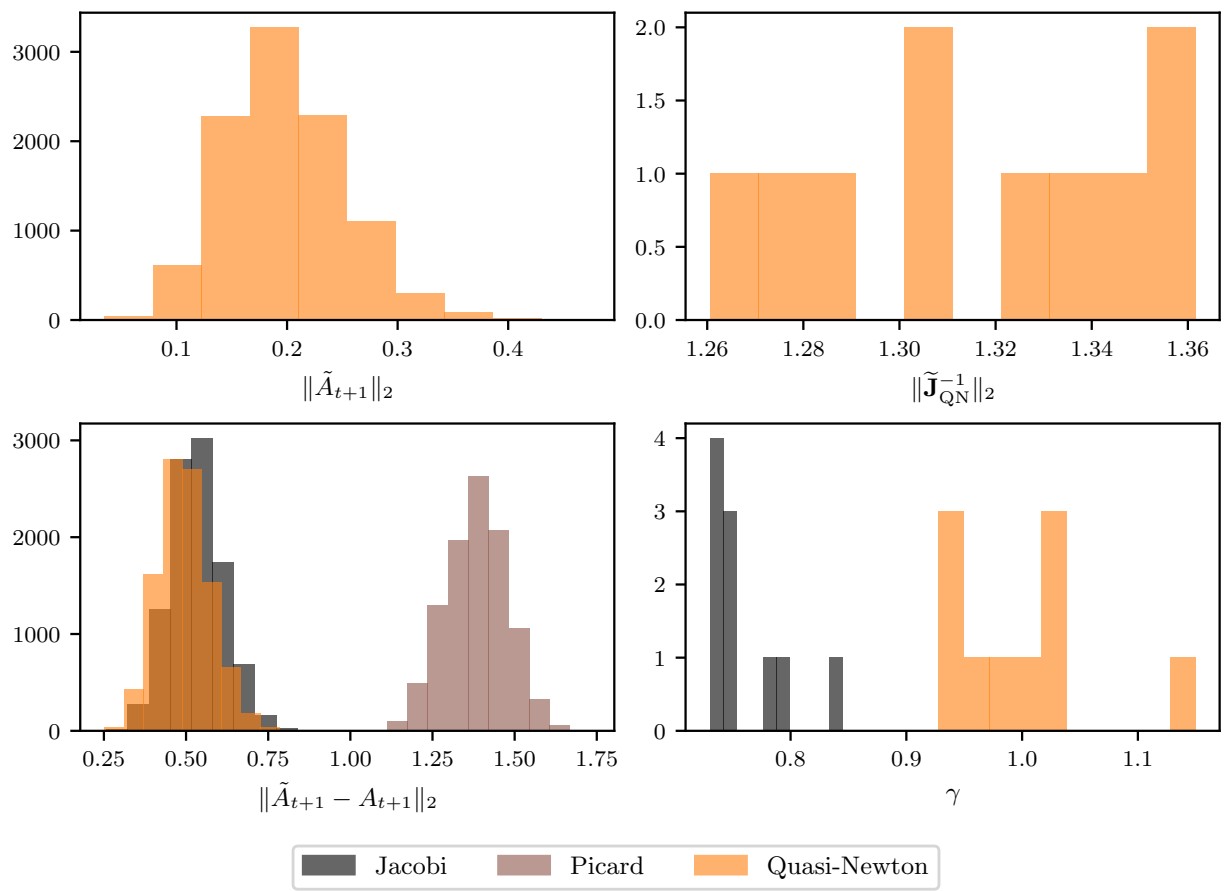

Figure 8: **Understanding the convergence rates in Figure 3.** In the setting of the GRU experiment for $D = 8$ and $T = 1000$, we plot relevant quantities for understanding the convergence rates of different methods over 10 random seeds. **(Top left.)** For all $T = 1000$ time steps and over all 10 random seeds, we plot the spectral norm of the approximate Jacobian for the quasi-Newton iterations we consider in this paper, i.e. $\mathrm{diag}[\partial f_{t+1}/\partial x_t(x_t^\star)]$. **(Top right.)** For each of the 10 random seeds, we plot $\|\widetilde{\mathbf{J}}_{\mathrm{QN}}(\mathbf{x}_{1:T}^\star)^{-1}\|_2$. We observe that they are always larger than one. **(Bottom left.)** We plot the difference between approximate Jacobians and true dynamics Jacobians over all time steps and seeds for quasi-Newton, Jacobi, and Picard iterations. We observe that this difference for Picard iterations is always larger than one, and so we would intuitively expect Picard iteration to be very slow for parallelizing GRUs. This behavior is precisely what we see in Figure 3. **(Bottom right.)** Across the 10 random seeds, we plot the value of $\gamma$ for Jacobi and quasi-Newton iterations (Picard would be $O(T)$ and so is not shown). Because $\|\widetilde{\mathbf{J}}_{\mathrm{J}}(\mathbf{x}_{1:T}^\star)^{-1}\|_2 = 1$, the 10 $\gamma_J$'s are equivalent to the maximum values from the differences in (bottom left) over the 10 random seeds. However, since (top right) shows that $\|\widetilde{\mathbf{J}}_{\mathrm{QN}}(\mathbf{x}_{1:T}^\star)^{-1}\|_2 > 1$, we observe that the values of $\gamma_{QN}$ are larger than the values of $\gamma_{\mathrm{J}}$ in (bottom right). In summary, because the values of $\gamma_J$ are smaller than the values of $\gamma_{\mathrm{QN}}$, we would intuitively expect Jacobi to converge in fewer fixed-point iterations, which is exactly what we observe in Figure 3.

We observe qualitatively similar results to the experiment shown in Figure 4: viz, that Picard and quasi-Newton iterations enjoy similar convergence rates and wall-clock time in settings where the Jacobian is well-approximated by the identity matrix.

Finally, we include an experiment designed to show how the density of the Jacobian favors Newton iterations over Picard iterations. In the experiment depicted in Figure 10, we again parallelize discretized Langevin

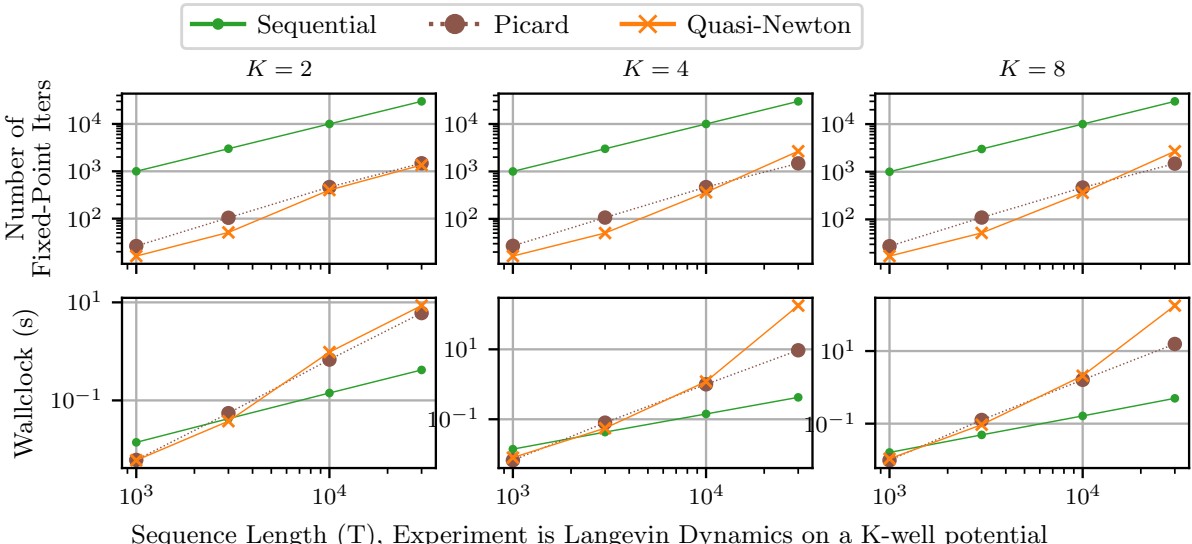

Figure 9: In a similar setting as Figure 4, we instead consider evaluating Langevin dynamics on a potential defined by the negative log probability of a mixture of $K$ anisotropic Gaussians. Here we keep $D = 128$ throughout. We observe qualitatively similar behavior to that shown in Figure 4, where both Picard and quasi-Newton iterations enjoy similar convergence and wall-clock speed.

diffusion two-well potentials. However, this time we use three different discretization step-sizes $\epsilon$: we use $1 \times 10^{-5}, 1 \times 10^{-4}$, and $1 \times 10^{-3}$. We showed that for step size $\epsilon = 1 \times 10^{-5}$ (the smallest considered, Jacobian most diagonal) that Newton and Picard both converged in a small number of fixed-point iterations, with Picard therefore running faster overall. In contrast, for $\epsilon = 1 \times 10^{-3}$ (the largest considered, Jacobian closer to dense), Newton continued to converge in a small number of iterations whereas Picard required considerably more iterations to converge. In this setting, Newton remained faster than sequential evaluation on wallclock time, whereas Picard did not.

## E.5 Implementation Considerations

In this section, we note that while the presented fixed-point methods are parallelizable, their real-world efficiency depends on the compute environment. Choosing the appropriate method requires balancing convergence speed, computational intensity, and resource availability.

**Parallel Associative Scan is Hardware-Sensitive** As we have shown, the fixed-point methods discussed can be cast as LDSs and therefore be parallelized over the sequence length using a parallel associative scan. However, the practical performance of these operations depends strongly on the hardware and low-level implementation details.

For example, modern GPUs (e.g., NVIDIA A100, H100) are highly optimized for tensor operations and large batch matrix multiplications, which favor methods like Newton and quasi-Newton iterations that perform fewer, more intensive steps.

The performance gains from using more iterations of lighter-weight updates (e.g., Picard iterations) are hardware-dependent. For example, Sarnthein (2025) showed that writing custom kernels for linear recurrences as tensor operations can yield near-optimal memory-bound performance.

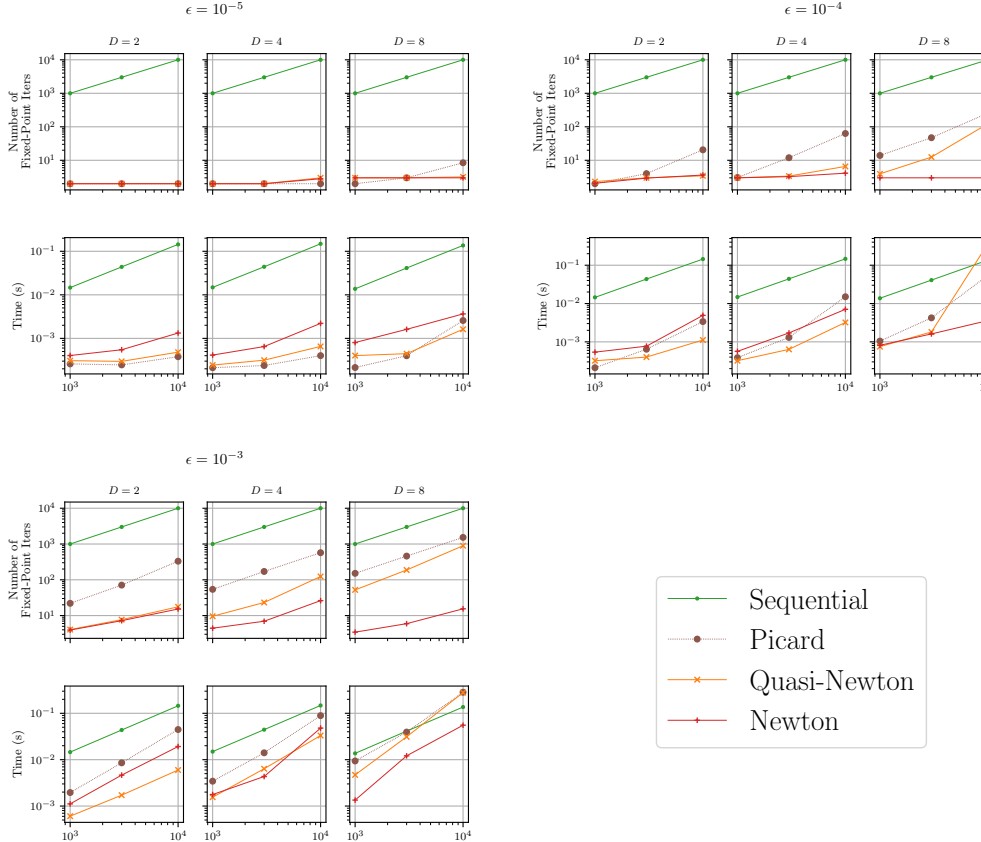

Figure 10: **Varying the step size** $\epsilon$. In this variant of discretized Langevin diffusion on a two-well potential, we vary the step size $\epsilon$. For the smallest step size $\epsilon = 1 \times 10^{-5}$ the Jacobian is most diagonal dominant of these three settings, and Picard iterations are faster on a wallclock time than Newton iterations. For the largest step size $\epsilon = 1 \times 10^{-3}$ the Jacobian is least diagonal dominant of these three settings, and Picard iterations are slower on a wallclock time than Newton iterations. We do not show Jacobi iterations because they already were demonstrated to struggle in discretizations of Langevin diffusions in Figure 4.

**Memory Usage and Tradeoffs** The memory requirements of different fixed-point iterations vary substantially. In particular:

- **Newton iterations** require storing and manipulating full Jacobians, which scales as $\mathcal{O}(D^2T)$ in space. This can become prohibitive for long sequences or high-dimensional hidden states.

- **Quasi-Newton iterations** reduce memory cost by using diagonal Jacobians, bringing the complexity down to $\mathcal{O}(DT)$. This is often a sweet spot for balancing memory usage and convergence rate.

- **Picard and Jacobi iterations** are the most memory-efficient, requiring only the storage of current and previous state estimates ($\mathcal{O}(DT)$), and no Jacobian-related storage.

In practice, when parallelizing over long sequences ($T \gg D$), the memory cost is often dominated by the size of intermediate state representations and the need to unroll computations over multiple fixed-point iterations. Chunking (dividing the sequence into smaller windows) and truncation (limiting the number of fixed-point iterations) are useful strategies to reduce memory usage in these settings (Dao et al., 2022; Shih et al., 2023; Selvam et al., 2024; Geiping et al., 2025; Zoltowski et al., 2025)

**Numerical Stability** For all fixed-point methods, numerical stability is a concern (Yaghoobi et al., 2025). In particular, LDS matrices with spectral norm close to or greater than one can cause numerical instabilities in the parallel scan operation (Gonzalez et al., 2024; 2025). This is especially critical in high-precision tasks or long sequences, and practitioners should monitor for numerical divergence or the accumulation of floating-point error.

# F Extended related work

## F.1 Discussion of parallel chord methods

Ortega & Rheinboldt (1970) discuss at length iterative methods for solving arbitrary systems of nonlinear equations $\mathbf{F}(\mathbf{x}) = 0$ using iterations of the form

$$\mathbf{x}^{(i+1)} = \mathbf{x}^{(i)} - \widetilde{\mathbf{J}}(\mathbf{x}^{(i)})^{-1}\mathbf{F}(\mathbf{x}^{(i)}) \tag{21}$$

for some matrix $\widetilde{\mathbf{J}}(\mathbf{x}^{(i)})$. In general, $\widetilde{\mathbf{J}}$ can be a function of the current iterate $\mathbf{x}^{(i)}$ or a fixed and constant matrix. Newton's method corresponds to

$$\widetilde{\mathbf{J}}(\mathbf{x}^{(i)}) = \frac{\partial \mathbf{F}}{\partial \mathbf{x}}(\mathbf{x}^{(i)}).$$

When $\widetilde{\mathbf{J}}$ is fixed and constant, Ortega & Rheinboldt (1970) describe the resulting family of fixed-point iterations as *parallel-chord methods*. However, we will use this term for *all* iterative methods with updates of the form in eq. (21), which includes both Newton and Picard iterations.

The term "parallel" in this context does not have to do with applying a parallel scan over the sequence length (which we discuss at length in this paper). Instead, "parallel" in "parallel-chord methods" refers to the way in which Newton's method finds the zero of a function by making a guess for the zero, and then forming a chord that is parallel to the function at the current guess (see Figure 11). In one-dimension, the linearization is a line (a chord), while in higher-dimensions the linearization is in general a hyperplane. In Newton's method, the chord/hyperplane is tangent to the function at the current guess, while for other parallel-chord methods the approximate linearization is in general not tangent.

The equation $\mathbf{F}(\mathbf{x}) = \mathbf{0}$ is a fully general way to represent a system of nonlinear equations. However, in this paper, we focus on parallelizing Markovian state space models. Thus, we consider the special setting where

$$\mathbf{F}(\mathbf{x}) = \mathbf{r}(\mathbf{x}_{1:T}) := [x_1 - f(x_0), x_2 - f(x_1), x_3 - f(x_2), \ldots, x_T - f(x_{T-1})], \tag{22}$$

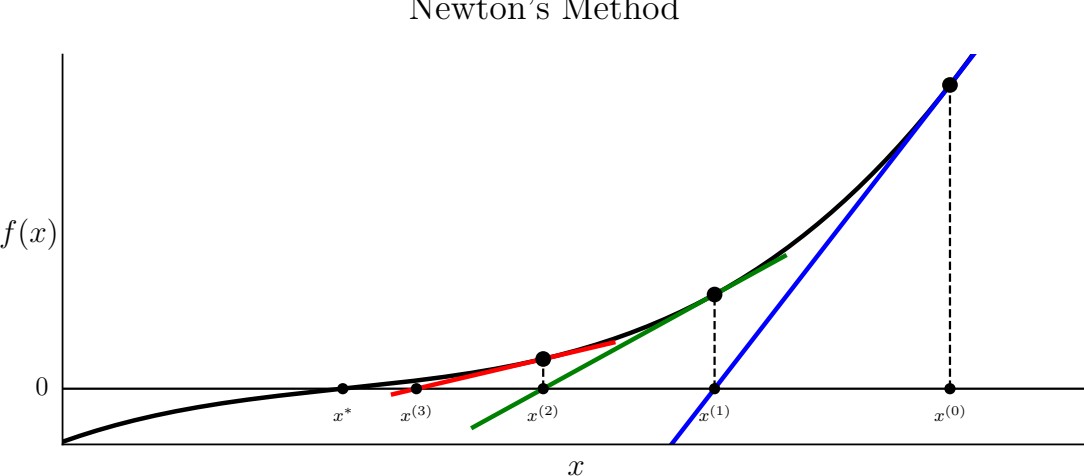

Figure 11: **The term "parallel" in parallel-chord methods.** Here we illustrate 3 iterations of Newton's method for root-finding on the one-dimensional cubic function $f(x) = (x - 0.4)^3 + 0.45(x - 0.4)$. We observe that each iteration of Newton's method involves forming a "parallel chord" to the function (shown in color).

where $f$ is the nonlinear transition function, as defined in eq. (1), and $\mathbf{r}$ is the residual function defined in eq. (10). Thus, in the context of parallelizing sequences, it follows that

$$\frac{\partial \mathbf{F}}{\partial \mathbf{x}}(\mathbf{x}) = \begin{pmatrix} I_D & 0 & 0 & \dots & 0 & 0 \\ -\frac{\partial f_2}{\partial x}(x_1) & I_D & 0 & \dots & 0 & 0 \\ 0 & -\frac{\partial f_3}{\partial x}(x_2) & I_D & \dots & 0 & 0 \\ \vdots & \vdots & \vdots & \ddots & \vdots & \vdots \\ 0 & 0 & 0 & \dots & I_D & 0 \\ 0 & 0 & 0 & \dots & -\frac{\partial f_T}{\partial x}(x_{T-1}) & I_D \end{pmatrix}. \tag{23}$$

When we plug this form of the Jacobian into $\widetilde{\mathbf{J}}$ in eq. (21) and simplify, we obtain the linear dynamical system in eq. (5), i.e. Newton's method.

In their treatment of Picard iterations, Ortega & Rheinboldt (1970) consider a more general formulation than that presented in Shih et al. (2023) or in eq. (8). Instead, similar to the definition presented in Appendix C.2.3 of Gu et al. (2021), Ortega & Rheinboldt (1970) define Picard iterations in the setting where we have removed a linear component of $\mathbf{F}$, namely we have written

$$\mathbf{F}(\mathbf{x}) =: \widetilde{\mathbf{J}}\mathbf{x} - \mathbf{G}(\mathbf{x}), \tag{24}$$

for some constant, nonsingular matrix $\widetilde{\mathbf{J}}$ and nonlinear function $\mathbf{G}(\cdot)$. Note that such redefinition of $\mathbf{F}(\cdot)$ in terms of $\widetilde{\mathbf{J}}$ and $\mathbf{G}(\cdot)$ is always possible and not uniquely determined. After making such a redefinition, Ortega & Rheinboldt (1970) define a Picard iteration as an update of the form

$$\mathbf{x}^{(i+1)} = \widetilde{\mathbf{J}}^{-1}\mathbf{G}(\mathbf{x}^{(i)}). \tag{25}$$

However, by multiplying both sides of eq. (24) by $\widetilde{\mathbf{J}}^{-1}$, it follows that

$$\widetilde{\mathbf{J}}^{-1}\mathbf{G}(\mathbf{x}^{(i)}) = \mathbf{x}^{(i)} - \widetilde{\mathbf{J}}^{-1}\mathbf{F}(\mathbf{x}^{(i)}),$$

showing that the Picard iterations as defined in eq. (25) fit into the parallel-chord framework set out in eq. (21). Note that Picard iterations as defined by Shih et al. (2023) or in eq. (8) of this paper also fit into

the framework of eq. (24): in the context of evaluating discretized ODEs, the residual defined in eq. (22) becomes

$$F_{t+1}(\mathbf{x}) = x_{t+1} - x_t - \epsilon g_t(x_t).$$

Thus, in the context of eq. (24), we have that the resulting $G_t(\mathbf{x}) = \epsilon g_{t-1}(x_{t-1})$, while the resulting $\widetilde{\mathbf{J}}$ operator is given by eq. (20).

When we plug this $\widetilde{\mathbf{J}}$ into eq. (21) and simplify, we obtain the linear dynamical system in the "Picard" row of Table 1. In general, the fixed-point methods of the common form given by eq. (4) all give rise to $\widetilde{\mathbf{J}} \in \mathbb{R}^{TD \times TD}$ matrices of the form shown in eq. (19).

Thus, Ortega & Rheinboldt (1970) unites Newton and Picard iterations for the general root finding problem $F(\mathbf{x}) = 0$ under the umbrella of parallel-chord methods, which are iterative updates of the form of eq. (21). The framework we provide in Table 1 can be understood as a specialization of parallel-chord methods for the particular problem of sequential evaluation discussed in eq. (1). Nonetheless, we focus on how in the specific problem of sequential evaluation, which is of great interest in many areas of machine learning, a wide variety of fixed-point methods become iterative application of LDSs, allowing them to be parallelized over the sequence length with an associative scan. This important perspective about parallelizability, which is of great interest in machine learning, is not discussed in Ortega & Rheinboldt (1970) because they are considering a more general problem.

Ortega & Rheinboldt (1970) also discuss in their Chapters 7 and 10 how the closeness of the "parallel chord" (in general and in higher dimensions, the "approximating hyperplane") to the true linearization of the function (Newton's method) affects the number of iterations needed for the parallel-chord method to converge. This analysis is directly analogous to our study of the effect of $\left\|\widetilde{\mathbf{J}}(\mathbf{x}_{1:T}) - \mathbf{J}(\mathbf{x}_{1:T})\right\|_2$ on the rate of convergence of fixed-point methods, see Lemma 1 In particular, in Chapter 10 of Ortega & Rheinboldt (1970), they consider the rates of convergence of fixed-point methods with updates taking the form of

$$\mathbf{x}^{(i+1)} = \mathbf{U}(\mathbf{x}^{(i)}), \tag{26}$$

for some function $\mathbf{U}(\cdot)$. Ortega & Rheinboldt (1970) use the name *one-step stationary methods* such fixed-point methods with updates with the form eq. (26).

For parallel-chord methods of the form given in eq. (21), it follows that

$$\mathbf{U}(\mathbf{x}^{(i)}) = \mathbf{x}^{(i)} - \widetilde{\mathbf{J}}(\mathbf{x}^{(i)})^{-1}\mathbf{F}(\mathbf{x}^{(i)}). \tag{27}$$

In particular, in their Chapters 7 and 10, Ortega & Rheinboldt (1970) introduce and study $\sigma(\mathbf{U}, \mathbf{F}, \mathbf{x}^\star)$, which determines the rate of convergence of iterative methods with updates of the form given by eq. (26) to the solution $\mathbf{x}^\star$ of $\mathbf{F}(\mathbf{x}) = \mathbf{0}$. They define $\sigma$ as

$$\sigma(\mathbf{U}, \mathbf{F}, \mathbf{x}^\star) := \rho\left(\frac{\partial \mathbf{U}}{\partial \mathbf{x}}(\mathbf{x}^\star)\right), \tag{28}$$

where $\rho(M)$ denotes the spectral radius of a matrix $M$.

In the context of parallel-chord methods where $\mathbf{U}(\cdot)$ is given by eq. (27), it follows that

$$\frac{\partial \mathbf{U}}{\partial \mathbf{x}}(\mathbf{x}^\star) = \mathbf{I} - \widetilde{\mathbf{J}}(\mathbf{x}^\star)^{-1}\frac{\partial \mathbf{F}}{\partial \mathbf{x}}(\mathbf{x}^\star),$$

because $\mathbf{F}(\mathbf{x}^\star) = 0$. Thus it follows that if $\widetilde{\mathbf{J}} = \partial \mathbf{F}/\partial \mathbf{x}(\mathbf{x}^\star)$, then $\sigma = 0$. Thus, lower values of $\sigma$ indicate that $\widetilde{\mathbf{J}}$ is a good approximation of the Jacobian matrix $\partial \mathbf{F}/\partial \mathbf{x}$ evaluated at the zero $\mathbf{x}^\star$ of $\mathbf{F}$, while higher values of $\sigma$ indicate that $\widetilde{\mathbf{J}}$ is a poor approximation for $\partial \mathbf{F}/\partial \mathbf{x}$. Ortega & Rheinboldt (1970) then use $\sigma$ in their Chapter 10 (in particular, their Theorem 10.1.4) to prove linear rates of convergence[10] for one-step stationary methods within a neighborhood of the solution $\mathbf{x}^\star$.

---

[10]where the rate is given by $\sigma$

Thus, a takeaway from Ortega & Rheinboldt (1970) (as paraphrased from Gasilov et al. (1981)) is that the closer $\widetilde{\mathbf{J}}$ is to $\partial \mathbf{F}/\partial \mathbf{x}$, the fewer iterations are needed for convergence to $\mathbf{x}^\star$. This takeaway is extremely similar to our guidance, though we specialize to the particular system of equations given by eq. (3) that results from the goal of rolling out the Markov process given by eq. (1).

However, in the setting we consider in this paper—using fixed-point iterations of the form eq. (4) to solve nonlinear equations of the form eq. (3)—Theorem 10.1.4 of Ortega & Rheinboldt (1970) is actually *trivial*. By "trivial," we mean that it does not distinguish between the convergence rates of any of the fixed-point iterations we focus on in this paper.

To make this point more precisely, we review[11] the notion of *root-convergence*, more commonly known as *R-convergence*.

**Definition 3** (*R*-convergence). *Let $\mathcal{A}$ be a fixed-point operator with fixed-point $\mathbf{x}^\star$. Let $C(\mathcal{A}, \mathbf{x}^\star)$ be the set of all sequences generated by $\mathcal{A}$ which converge to $\mathbf{x}^\star$. Then the $R_1$-factors of $\mathcal{A}$ at $\mathbf{x}^\star$ are given by*

$$R_1(\mathcal{A}, \mathbf{x}^\star) := \sup \left\{ \limsup_{i \to \infty} \|\mathbf{x}^{(i)} - \mathbf{x}^\star\|^{1/i} \,\middle|\, \{\mathbf{x}^{(i)}\}_{i \geq 0} \in C(\mathcal{A}, \mathbf{x}^\star) \right\}. \tag{29}$$

Intuitively, $R_1(\mathcal{A}, \mathbf{x}^\star)$ gives the rate of linear convergence of a fixed-point operator $\mathcal{A}$ to its fixed-point $\mathbf{x}^\star$. Theorem 10.1.4 of Ortega & Rheinboldt (1970) implies that if $\mathcal{A}$ is a one-step stationary method with update given by $\mathbf{U}(\cdot)$, then $R_1(\mathcal{A}, \mathbf{x}^\star) = \sigma(\mathbf{U}, \mathbf{F}, \mathbf{x}^\star)$. Therefore, if $\sigma > 0$, then $\sigma$ is the rate of $R$-linear convergence of $\mathcal{A}$ to $\mathbf{x}^\star$, while if $\sigma = 0$, we say that $\mathcal{A}$ converges *R-superlinearly*. However, it is important to note that these definitions are *asymptotic* in nature.

The fixed-point iterations considered in this paper, i.e. following the common form eq. (4), all have $\sigma = 0$, and therefore can be said to converge $R$-superlinearly.

**Proposition 5.** *Let $\mathbf{F}(\mathbf{x}) = \mathbf{0}$ be a nonlinear equation of the form eq. (3) with solution $\mathbf{x}^\star$. Let $\mathcal{A}$ be a parallel-chord method with fixed-point $\mathbf{x}^\star$. Then*

$$\sigma\left(\mathbf{U}, \mathbf{F}, \mathbf{x}^\star\right) = 0.$$

*Proof.* Both $\partial \mathbf{F}/\partial \mathbf{x}(\mathbf{x}^\star)$ and $\widetilde{\mathbf{J}}(\mathbf{x}^\star)$ are lower-triangular matrices with all $D \times D$ identity matrices on their main block-diagonal. In particular, $\widetilde{\mathbf{J}}^{-1}$ is also a lower-triangular matrix with all $D \times D$ identity matrices on its main block-diagonal. Consequently, the product $\widetilde{\mathbf{J}}^{-1} \dfrac{\partial \mathbf{F}}{\partial \mathbf{x}}$ is also a lower-triangular matrix with all $D \times D$ identity matrices on its main block-diagonal. As a result, $\mathbf{I} - \widetilde{\mathbf{J}}^{-1} \dfrac{\partial \mathbf{F}}{\partial \mathbf{x}}$ is a lower-triangular matrix with all zeros on its main block-diagonal, and so has all its eigenvalues equal to 0. Consequently, its spectral radius is equal to zero. □

It may seem counterintuitive that even Jacobi iterations technically enjoy $R$-superlinear convergence in the context of parallelizing Markov processes. However, this seemingly strange result stems from the asymptotic nature of Definition 3 of $R$-convergence, and the fact that Proposition 1 of Gonzalez et al. (2024) guarantees that all fixed-point iterations of the form given by eq. (4) will converge to $\mathbf{x}^\star$ in a finite number of iterations ($T$, to be exact). Therefore, for any LDS fixed-point scheme, we always have $\lim_{i \to \infty} \|\mathbf{x}^{(i)} - \mathbf{x}^\star\| = 0$.

However, in both Proposition 4 of Lu et al. (2025) and Proposition 3 of this paper, we effectively get around this difficulty by considering the *spectral norm* instead of the *spectral radius*. The spectral norm always bounds the spectral radius, and so by focusing on spectral radius, Ortega & Rheinboldt (1970) could get tighter bounds (faster rates of convergence). However, in our setting the spectral radius cannot distinguish between any of the fixed-point methods, and so we instead use the looser bound provided by the spectral norm, which can distinguish between the different fixed-point methods. Note that the core entities are effectively the same, as $\gamma$ defined in eq. (15) is equal to $\|\partial \mathbf{U}/\partial \mathbf{x}(\mathbf{x}^\star)\|_2$.

---

[11] We follow the presentation of Chapter 9 of Ortega & Rheinboldt (1970), in particular Definition 9.2.1.

Finally, again, because all of our fixed-point methods converge in at most $T$ iterations, asymptotic notions of linear convergence are not suitable to fully capture the behavior of these fixed point methods. For this reason, we use empirical case studies in Section 4 to show the efficacy of the intuition, inspired by Proposition 3, that the closeness of $\tilde{A}_t$ to $\partial f_t / \partial x$ impacts the number of iterations needed for $\mathcal{A}$ to converge. This empirical approach also highlights how the increased computational cost of higher-order fixed-point methods affects wall-clock time on GPUs.

### F.2    Trust-region fixed-point methods

Many other fixed-point techniques can fit into the general framework we propose in Table 1 and eq. (4) (see the general algorithm in Algorithm 1). We focus on Newton, quasi-Newton, Picard, and Jacobi in the main text because of their prominence and canonicity. However, other fixed-point iterations also fit into this framework.

For example, Gonzalez et al. (2024) introduces the scale-ELK algorithm which for some $k \in [0, 1]$ sets the transition matrix $\tilde{A}_t$ to be

$$\tilde{A}_t = (1 - k)\frac{\partial f_{t+1}}{\partial x}.$$

Scale-ELK can also be applied to the diagonal approximation for quasi-Newton methods. However, scale-ELK introduces an additional hyperparameter $k$.

Therefore, we propose *clip-ELK*, which is a hyperparameter-free approach to achieve the same goal of a stable LDS. Clip-ELK applies to the diagonal approximation only, and simply clips each element of $\tilde{A}_t$ (which in this setting is a diagonal matrix) to be between $[-1, 1]$. Clip-ELK is immediately also a part of the framework set out in Table 1 of fixed-point methods that parallelize the evaluation of sequences via iterative application of LDS; moreover, it too must also converge globally by Proposition 1 of Gonzalez et al. (2024). An interesting and important direction for future work includes both developing and interpreting more fixed-point methods in the context of this framework that we propose.

