# OpenReview forum: "A Unifying Framework for Parallelizing Sequential Models with Linear Dynamical Systems"
_TMLR — Accepted by TMLR_

### Review · Reviewer_Hz5N · 2025-11-03

**Summary Of Contributions:**

The paper discusses techniques to parallelize sequential models. The authors propose a general framework based on linear dynamical systems (LDS) that unifies several recent fixed-point methods that parallelize sequential models, including Newton, Picard, and Jacobi iterations. A parallel scan algorithm is then used to solve the dynamical system in log time. Case studies on solving the group word problem, RNN’s, and Langevin diffusions are given to highlight the differences between the three iteration methods.

Strengths:

1. The proposed LDS framework is elegant and effectively captures the three discussed iteration methods. Clear derivations of the correspondences between the proposed framework and the three methods are also given.

2. The paper verifies the methods empirically and conducted ablation studies between different methods in different experiment settings. Through the experiments, the paper also gives an empirical rule for choosing an iteration method.

Weaknesses or Suggestions:

1. The contributions are limited. The Newton, Picard, and Jacobi iterations are all previous works, and turning these methods into the form of $x_{t+1}^{(i+1)} = f_{t+1}\left(x_t^{(i)}\right) + A_{t+1}\left(x_t^{(i+1)}-x_t^{(i)}\right)$ is rather straightforward.

2. The main heuristic for choosing between different iteration methods is based on the Jacobian $\frac{\partial f_{t+1}}{\partial x_t}$, where the Newton iteration uses the full Jacobian for the $A$ matrix. While the proposed framework uses a general $A$ matrix, the discussion in the paper seems to be still centered around the Newton method. Is it possible to discuss a whole new family of iteration methods based on the general framework and analyze the new family both theoretically and empirically?

3. Is it possible to give an analytical bound for the number of fixed point iterations based on some metric of distance between $A_{t+1}$ and the Jacobian $\frac{\partial f_{t+1}}{\partial x_t}$?

4. The experiments are mostly synthetic. It would be stronger to show speed-up results on trained real tasks, e.g. a real diffusion sampling with SDE or ODE samplers.

5. Although an introduction of the parallel scan algorithm is given in the appendix, it would be better to outline the exact parallel scan algorithm to solve the LDS’s mentioned in the main paper.

**Audience:**

Yes

**Audience Explanation:**

Since sequential sampling is prevail in LLMs, developing efficient parallel sampling techniques is definitely valuable and of broad interest to the community.

**Claims And Evidence:**

Yes

**Claims Explanation:**

The claims made are generally supported by evidence. However, as mentioned above, since the experiments are mostly synthetic, it would be more convincing to include more real-world experiments with larger scales.

**Requested Changes:**

Please see my suggestions in the Summary Of Contributions section.

---

> ### Author Response · Authors · 2025-11-24
>
> Thank you for your thoughtful feedback! We appreciate that you found our framework to be elegant, effective, and clear. We now discuss your suggestions. We have uploaded a revised pdf with changes tracked in red.
>
> > 1. Limited contributions
>
> We agree that our main contribution is the unifying framework and not any fixed point iteration in particular. However, while the derivations are clear and straightforward once they are shown, to the best of our knowledge these connections were not broadly known in the machine learning community. This lack of common language has led to duplication of work. Our paper, by being clear and easy-to-follow, will help in the sharing of knowledge across domains.
>
> > 2. Develop a new family
>
> We agree that future development of fixed point iterations is a very interesting question. We tried to discuss this in our original submission, but have now stated this explicitly in “Future directions” (see page 12). The fact that this paper opens up such interesting directions for future directions is one of the reasons why we believe the community would benefit from this publication. However, the scope of this technical note is simply introducing this unifying framework. We leave further extensions based on this unifying framework to future work.
>
> > 3. Analytical bounds
>
> We agree that analytical bounds are a very interesting direction for future work. We tried to make this point clear in the submission, but will add this direction to a more explicitly denoted “future directions” paragraph (see page 12). We discussed how current literature is currently insufficient for proving such bounds. Getting an exact numerical formula is difficult: for example, Gonzalez & Kozachkov et al 2025 (https://arxiv.org/abs/2508.16817) only provide bounds on a constant important for the rate of convergence. For this reason, we envision work on this question as part of a broader research program outside the scope of this paper. Nonetheless, we are excited that this work is prompting such interesting questions, which is why we think the community would benefit from publication of this work.
>
> > 4. Synthetic experiments
>
> We focus on synthetic experiments to highlight in controlled settings the features of dynamical systems that favor different fixed point methods. In fact, reviewer N66A specifically requested a controlled experiment to rigorously analyze the difference between the algorithms by setting the discretization time-step $\epsilon$ to various values to toggle the Jacobian matrix from more diagonalized to dense.
>
> We added precisely this experiment (See Figure 8). We showed that for step size $\epsilon=1e-5$ (the smallest considered, Jacobian most diagonal) that Newton and Picard both converged in a small number of fixed-point iterations, with Picard therefore running faster overall. In contrast, for $\epsilon=1e-3$ (the largest considered, Jacobian closer to dense), Newton continued to converge in a small number of iterations whereas Picard required considerably more iterations to converge. In this setting, Newton remained faster than sequential evaluation on wallclock time, whereas Picard did not. This experiment further corroborates our guidance as to how Jacobian structure affects the algorithmic choice.
>
> Moreover, all of these fixed-point methods have individually shown speed-ups on large scale tasks (see for example Danieli et al ‘25 [https://arxiv.org/abs/2510.21450], which uses Newton iterations for speed-ups when training 7B parameter RNNs for language modeling, or Shih et al ‘23 [https://arxiv.org/abs/2305.16317], which uses Picard iterations for speed-ups on StableDiffusion-v2). The goal of our paper, however, is to unify the different fixed-point iterations and show precisely where each excels, which can be done more clearly in the context of controlled synthetics.
>
> > 5. Parallel scan in main text
>
> We agree and have added (see page 3).

---

### Review · Reviewer_LPFg · 2025-11-05

**Summary Of Contributions:**

This paper studies the topic of solving the fixed-point problem of sequential processes in parallel. In particular, the paper investigates a unified underlying framework for Newton, Quasi-Newton, Picard, and Jacobi iterations. By contrasting the difference of the algorithms above under the same framework, the paper identifies conditions where each algorithm is most suitable for. The paper accompanied the observation with examples in the real-world applications, and supported the claim with experimental results.

**Audience:**

Yes

**Audience Explanation:**

The general problem of sequential process indeed appears in various machine learning settings, including unrolled networks, RNNs, and diffusion models. Speeding up this process can easy the computation during the inference of these models. The paper discusses the situations where different existing algorithms are most suitable for, which can be a guideline for practitioner to choose the correct algorithm.

**Broader Impact Concerns:**

None.

**Claims And Evidence:**

No

**Claims Explanation:**

First, the paper claims that the iterates of Newton, Quasi Newton, Picard, and Jacobi iterations for nonlinear recursions can be modeled as linear dynamical systems. However, in the formulation of Eq. (4), the iterate-wise update still depends on a nonlinear function $f_{t+1}(x_t^{(i)})$. It is not clear why Eq. (4) falls into the category of a linear dynamical system or why it can be solved using parallel scan algorithms. Moreover, the paper claims that how the structure of the problems informs which method one should use. The definition of the structure of the problem is vague here, as in the examples of Section 3, the paper simply discussed the dependency on the property of the per-iteration Jacobian. I believe this is a dependency more on the iterates rather than the general structure of the problem itself.

**Requested Changes:**

1. I suggest the paper to include a discussion of the parallel scan algorithm, and how it applies to the particular case of solving linear dynamical systems.


2. The paper should clarify the assumption of $f_{t+1}(x_t^{(i)})$ in Eq. (4), and, if it is a nonlinear function, explain why the resuling updates falls into the category of a linear dynamical systems.

3. The paper should be more specific on the "structure" of the problem instance, and explain that the choice of the algorithms depends on the per-iterate Jacobian matrix.

---

> ### Author Response · Authors · 2025-11-06
> **Explanations and evidence to clarify discussion**
>
> Thank you very much for your review and feedback! We appreciate your time and effort.
>
> We discuss your suggested changes and how we have incorporated them into our revised pdf (uploaded, with changes tracked in red).
>
> ## 1. “Include a discussion of parallel scan algorithm, and how it applies to solving LDS”
>
> We completely agree with your recommendation, which is why in our original submission we have Appendix A: The Parallel Scan: A gentle introduction. We explicitly discuss solving LDSs (see middle of page 17). We appreciate your recommendation that we move a discussion of parallel scans for LDSs into the main text of the paper (see page 3), and we have done so.
>
> ## 2. Clarify $f_{t+1}(x_t^{(i)})$ in Equation 4.
>
> This is a good point about clarification we need to make.
>
> As we show in Equation 2, the goal of recasting sequential evaluation as fixed-point iteration is to go from iteration $(i)$ with values $x_{1:T}^{(i)}$ to iteration $(i+1)$ with values $x_{1:T}^{(i+1)}$. During the $(i+1)$st iteration (i.e. when we go from $x_{1:T}^{(i)}$ to $x_{1:T}^{(i+1)}$), **we already know all the values of $x_{1:T}^{(i)}$.** So,  $f_{t+1}(x_t^{(i)})$ is **known in advance**, and its computation over the sequence length is embarrassingly parallel (it can be done with a map).
>
> We should have explicitly stated that Equation 4 is an LDS for $x^{(i+1)}$.
>
> Based on your feedback, we have added Equation 5, writing Equation 4 as
>
> $x_{t+1}^{(i+1)} = A_{t+1} x_t^{(i+1)} + (f_{t+1}(x_t^{(i)}) - A_{t+1} x_t^{(i)}).$
>
> This is an LDS of the form
>
> $x_{t+1}^{(i+1)} = A_{t+1} x_t^{(i+1)} + b_{t+1}$,
>
> where
>
> $b_{t+1}  := f_{t+1}(x_t^{(i)}) - A_{t+1} x_t^{(i)}$.
>
> Doing so more explicitly shows how each fixed-point iteration $(i+1)$ is a linear dynamical system for the (previously unknown) values of $x^{(i+1)}$. Notice that the “bias” term $b_{t+1}$ is a function only of $x^{(i)}$, which is already known (it’s our current guess for the trajectory).
>
> ## 3. “Structure” of problem instance
>
> Yes, this is a good point to clarify. By “structure” we mean the “structure” of the Jacobian of the dynamics, that is $\dfrac{\partial f}{\partial x}$. We did not mean to be vague about this, see the bottom of page 6 where we write “the structure of the Jacobian.” However, we have now highlighted that we mean “structure of the Jacobian matrix” when we first refer to “structure” at the end of Section 1 (“Introduction”).
>
> You are right that the value of Jacobian itself does depend on the iterate, i.e. in general $\dfrac{\partial f}{\partial x}(x_t^{(i)})$ need not equal $\dfrac{\partial f}{\partial x}(x_t^{(i+1)})$. However, in many problems, and in particular the problems we highlight in this paper, the *structure* of the Jacobian does *not* depend on the iterate. For example, in the group word problem, $A_t$ is *always* a permutation matrix (i.e., the structure is that of the permutation matrix). In the discretized Langevin diffusion, we derive (see Figure 3a) that the Jacobian is always of the form $I_D - \epsilon \nabla^2 \phi(x)$, where $\epsilon$ is extremely small (i.e., the structure is a matrix that is diagonally dominant and very close to the identity in Frobenius norm).
>
> Thank you again for your feedback! It has helped us clarify the exposition and strengthen the paper.

---

> > ### Comment · Reviewer_LPFg · 2025-12-01
> >
> > Thank you for the response and the improvements made for the paper. I find the discussion of the parallel scan algorithm and the explanation of why the iterates form a LDS clear and convincing.
> >
> > Regarding the "structure" of the problem instance, I still think it is a bit vague. Based on my background, I found the paper interesting because the sequence process appears in many deep learning models, and for the paper to be applied to deep learning, I recommend adding a section relating the Jacobian properties of neural networks to the results of the paper. If possible, adding experimental results outlining interesting common behavior of the Jacobian in the sequential process, and show its correlation with the best choice of the algorithm can be an interesting improvement. If the author find it hard to add the experiment in the limited time, I would suggest to add at least a discussion of how to extend the conclusion to the case where the Jacobian can be changing.

---

### Review · Reviewer_N66A · 2025-11-10

**Summary Of Contributions:**

## Contributions:

This paper proposes a new framework to unify four different algorithms for solving fixed-point iteration in the language of linear dynamical systems. The resulting LDS is clean, showing that the four different algorithms—Jacobi, Picard, Quasi-Newton, and Newton—differ only in the transition matrix in the iteration rule. The resulting practical guidance, "Use the simplest approximate Jacobian as possible, but no simpler," is clean and precisely captures the essence: to use as much full Jacobian information as possible at minimal cost.

## Strengths:

The strengths are clear; the proposed framework and summarization of ideas are clean and indeed provide a novel, unified view of these algorithms.

## Weaknesses:

Despite generally enjoying this work, I feel its depth is insufficient in both algorithmic and practical aspects.

Firstly, from an algorithmic aspect, the current version only discusses overly-simplified algorithms (Jacobi, Picard) or overly-expensive algorithms (diagonalized Newton and Newton). These algorithms, most of the time, are not optimal in practical scenarios. A majority of scenarios in machine learning involve a more general form of quasi-Newton method, where the transition matrix
is estimated in an economical way; for example, Anderson acceleration considers using history residuals to estimate the full Jacobian. Therefore, I think the current version misses a discussion on more general forms of quasi-Newton methods, which are more widely used and are more practical choices.

Secondly, from an experimental aspect, the depth is also not enough to support the claim "Use the simplest approximate Jacobian as possible, but no simpler." The current experiments use three different settings, where they have different Jacobian structures specific to their problem setting. However, this is not very instructive guidance, because most of the time a practitioner cannot know the Jacobian structure very well before running these algorithms, especially when the full Jacobian information is extremely hard to evaluate.

**Audience:**

Yes

**Audience Explanation:**

Yes, because fixed-point iteration is a powerful and ubiquitous modeling tool in the machine learning community.

**Broader Impact Concerns:**

This paper focuses on a unified algorithmic framework for fixed-point iteration, and I did not see any ethical issues.

**Claims And Evidence:**

Yes

**Claims Explanation:**

The main claim is that the authors propose a framework that unifies several classical fixed-point iteration algorithms, which is evident in the summarization in Table 1.

The experimental section (Section 3) also validates how the problem structure favors different algorithms.

**Requested Changes:**

Overall, I have three suggestions for the authors:

In Table 1 and the discussion, I suggest the authors include more practical forms of the quasi-Newton method, where the transition matrix is estimated rather than being exactly computed. This will make the framework cover more widely used and practical algorithms.

If possible, I suggest the authors discuss ways for practitioners to probe the black-boxed Jacobian structure before deciding on an algorithm. For example, via finite differences or other economical ways.

Thirdly, I suggest the authors use a more controlled experiment to rigorously demonstrate the differences between the algorithms. For example, in the Langevin diffusion section, you could set epsilon to various values to toggle the Jacobian matrix from more diagonalized to more dense, and then validate how the performance of different algorithms varies among these settings. This would provide a more precise view of how the Jacobian structure affects the algorithmic choice.

---

> ### Author Response · Authors · 2025-11-24
>
> Thank you for your thoughtful and helpful feedback. We have incorporated your feedback into our uploaded revised pdf (see changes in red). We discuss your three suggestions below
>
> > 1.) Estimating the transition matrix
>
> This is a good point. We have increased the emphasis on estimating the transition matrix in our discussion. This includes both the Hutchinson method for stochastically estimating the diagonal of the Jacobian matrix using a single function evaluation (which we use in all of our experiments). We also add a discussion of secant-based methods such as Triangular Anderson Acceleration (TAA), used by Tang et al ‘24 to parallelize sampling from diffusion models. See Table 1 and Section 2.2.
>
> > 2.) Probing the black-box Jacobian structure
>
> We now include a discussion of probing the black-box Jacobian structure using finite difference methods. See page 12.
>
> > 3.) Controlled experiment.
>
> This was a great suggestion.
>
> We liked your recommendation to use a  controlled experiment to rigorously analyze the difference between the algorithms by setting the discretization time-step $\epsilon$ to various values to toggle the Jacobian matrix from more diagonalized to dense.
>
> We added precisely this experiment (See Figure 8). We showed that for step size $\epsilon=1e-5$ (the smallest considered, Jacobian most diagonal) that Newton and Picard both converged in a small number of fixed-point iterations, with Picard therefore running faster overall. In contrast, for $\epsilon=1e-3$ (the largest considered, Jacobian closer to dense), Newton continued to converge in a small number of iterations whereas Picard required considerably more iterations to converge. In this setting, Newton remained faster than sequential evaluation on wallclock time, whereas Picard did not. This experiment further corroborates our guidance as to how Jacobian structure affects the algorithmic choice.
>
> Overall, thank you for your thoughtful feedback which has strengthened this paper!

---

### Author Response · Authors · 2025-11-24

We thank the reviewers for their time and helpful feedback!

Our paper presents a unifying framework for parallelizing nonlinear recursions with linear dynamical systems (LDSs). We show that major fixed-point iteration families like Jacobi, Picard, and Newton can all be written as LDSs, which can be parallelized over the sequence length with parallel scans. This contribution brings together advances from across disparate fields, including sequence modeling with recurrent neural networks (RNNS), sampling from diffusions, and more, unifying them in a common language to serve as a foundation for future work.

We are heartened by the positive response to our unifying framework. Reviewers described our proposed framework as “novel,” “clean,” “effective,” and “elegant.” We appreciate that all reviewers answered “Yes” to whether TMLR’s audience would be interested in knowing the findings of the paper. Two of the three reviewers also responded “Yes” to whether our claims are supported by convincing evidence. We believe the only reviewer who responded “No” did so because of a misunderstanding; we have used that feedback to further clarify and strengthen the submission.

## Revisions thanks to feedback

We also appreciate the helpful feedback from reviewers. We have incorporated this feedback to strengthen the paper. We have submitted a revised version of the paper, with all changes indicated in red. We highlight the major revisions below.

### Discussion of the parallel scan in main text

The parallel scan is a fundamental ingredient of our unifying framework, as it is a powerful primitive for parallelizing LDSs. For this reason, we wrote an extensive introduction to it in Appendix A. Nonetheless, two reviewers also recommended that we add a brief description in main text. We have done so.

### Clarification that each fixed point iteration is a linear dynamical system (LDS)

Reviewer LPFg was confused how Eq. (4) was an LDS. We added a further algebraic manipulation to provide in Eq. (5) a more standard depiction of an LDS, as

$ x_{t+1}^{(i+1)} = A_{t+1} x_t^{(i+1)} + b_{t+1}.$

Crucially, fixed point iteration $(i+1)$ takes us from $x^{(i)}$ (already known for all time) to $x^{(i+1)}$. The dynamics matrix $A_{t+1}$ and bias $b_{t+1}$ can be written as functions of $x_t^{(i)}$, which is already known.

### Strengthened experiments

Two reviewers asked for additional experiments to further demonstrate our unifying framework.

We particularly appreciated the recommendation from Reviewer N66A to use a controlled experiment to rigorously analyze the difference between the algorithms, for example setting the discretization time-step $\epsilon$ to various values to toggle the Jacobian matrix from more diagonalized to dense.

We added precisely this experiment. We showed that for step size $\epsilon=1e-5$ (the smallest considered, Jacobian most diagonal) that Newton and Picard both converged in a small number of fixed-point iterations, with Picard therefore running faster overall. In contrast, for $\epsilon=1e-3$ (the largest considered, Jacobian closer to dense), Newton continued to converge in a small number of iterations whereas Picard required considerably more iterations to converge. In this setting, Newton remained faster than sequential evaluation on wallclock time, whereas Picard did not. This experiment further corroborates our guidance as to how Jacobian structure affects the algorithmic choice

## Conclusion

Thanks to the reviewer feedback, we have strengthened the paper with clearer exposition and additional experiments that further buttress our claims. Our unifying framework is correct and applicable to many settings of interest. We hope that its publication in TMLR will serve the community by providing a common language for understanding fixed-point iterations for parallelizing nonlinear recursions across many fields.

---

### Decision · Action_Editor_zuT6 · 2025-12-20

**Recommendation:** Accept as is

**Additional Comments:**

Parallelization over the (long) sequence length has drawn a lot of attention lately. This work presented a unified framework to cover some of the most popular methods in this direction. The key idea is to reformulate a sequential evaluation as a fixed-point in the product space, and a subsequent linearization step allows the application of parallel scan (e.g., divide-and-conquer for matrix multiplication). This paper is expertly written, with convincing details and reasonable experimental illustrations. In the revision, the authors added further explanation that improved readability (for more casual readers). All reviewers found the proposed unified framework interesting while some expressed reservation on its significance. Nevertheless, by abstracting away some of the more tedious details, this work provides a clean and nice introduction to the core ideas and could be very useful for the broader audience. In the end all reviewers recommended acceptance, which I concur.

One small nitpicking on the reference:
- Mónika Farsang, Ramin Hasani, Daniela Rus, and Radu Grosu. Scaling up liquid-resistance liquidcapacitance networks for efficient sequence modeling. NeurIPS, 2025.

The list of authors does not match here: https://openreview.net/forum?id=ENYvdnyhLl (see also https://neurips.cc/virtual/2025/loc/san-diego/poster/119140), possibly due to a confusing arxiv version (https://arxiv.org/pdf/2505.21717v6) and a NeurIPS workshop version (https://neurips.cc/virtual/2025/loc/san-diego/133803).

**Audience:**

Yes

**Audience Explanation:**

Parallelization over the sequence length has drawn a lot of recent attention in nonlinear RNNs and state-space models. This work provides a nice introduction to this recent topic that could be interesting to many practitioners.

**Claims And Evidence:**

Yes

**Claims Explanation:**

The unified framework is shown analytically to capture many existing work as special cases, and numerical simulations (on three applications) were performed to illustrate the algorithmic variants.